# A Contrastive Teacher-Student Framework for Novelty Detection under Style Shifts

## Abstract

There have been several efforts to improve Novelty Detection (ND) performance. However, ND methods often suffer significant performance drops under minor distribution shifts caused by changes in the environment, known as style shifts. This challenge arises from the ND setup, where the absence of out-of-distribution (OOD) samples during training causes the detector to be biased toward the dominant style features in the in-distribution (ID) data. As a result, the model mistakenly learns to correlate style with core features, using this shortcut for detection. Robust ND is crucial for real-world applications like autonomous driving and medical imaging, where test samples may have different styles than the training data. Motivated by this, we propose a robust ND method that crafts an auxiliary OOD set with style features similar to the ID set but with different core features. Then, a task-based knowledge distillation strategy is utilized to distinguish core features from style features and help our model rely on core features for discriminating crafted OOD and ID sets. We verified the effectiveness of our method through extensive experimental evaluations on several datasets, including synthetic and real-world benchmarks, against nine different ND methods.

## 1 Introduction

Novelty detection (ND) has emerged as a critical component in developing reliable real-world machine learning models. The primary task of ND is to distinguish Out-of-distribution (OOD) samples from the in-distribution (ID) samples during inference, using only unlabeled ID samples for training (5; 67; 74; 96). This task is essential across various computer vision applications, including industrial defect detection, medical disease screening, and video surveillance (43; 72; 94; 74). However, these methods often experience significant performance drops when confronted with test data exhibiting minor distribution shifts in their *style*, such as changes in the test sets due to environmental variations (See Figure 1) (13; 11; 85; 86; 20).

A robust detector should be invariant to changes in the style features, as variations in these features do not change a sample's label (ID or OOD). Instead, it should be expected to learn the core features which determine the label (3; 58; 13). Robustness against style shifts is a crucial aspect of ND methods since variations in style are common in real-world applications. For instance, an ND method for autonomous driving tasks trained on images from Germany streets (21) should also perform effectively on the streets of Los Angeles (71), despite variations in style features caused by different lighting and atmospheric conditions. A similar challenge exists in medical imaging, where shifts can occur due to different imaging equipment, patient positioning, and variations in tissue properties (95).

The vulnerability of existing ND methods stems from their implicit assumption that the training data should strongly mirror the test data, even in stylistic features. This leads to the misprediction of an ID test sample with a different style feature as OOD. Furthermore, training data in the ND setup is limited to ID samples. By relying solely on ID samples, the detector learns a correlation between the dominant style features present in ID samples and the label. Consequently, the detector mistakenly uses these style features for discrimination instead of focusing on core features. As a result, the detector incorrectly predicts an ID test sample with a different style as OOD and an OOD sample with a similar style as ID (13).

Notably, current domain generalization and domain adaptation methods cannot be applied to develop robust ND methods against distribution shifts, as they require access to labeled training data or extra

data from different environments, which are not available in the ND setup (40; 68; 99; 103; 63; 109; 105). Furthermore, our study distinguishes itself from recent works such as RedPanda (20) and PCIR (13), which leverage different environmental annotations as additional information to improve the ND robustness. In many real-world scenarios, ID training samples are collected from unknown environments, and hence such metadata is often missing (11; 77).

Motivated by these challenges, we propose crafting an auxiliary OOD set by identifying the core features of the ID samples and distorting them. To identify the core features, we employ a feature attribution method (Grad-CAM (80)) applied on the output of a pre-trained network when fed with the ID samples. We apply light augmentations (e.g., color jitter (16; 35; 36)) to the input, and compute saliency maps for both the original and augmented versions. By taking the element-wise product of these saliency maps, we derive a final saliency map where higher values correspond to the core features of the assumed ID sample. These light augmentations facilitate producing a final saliency map agnostic to style shifts. Subsequently, hard transformations (52; 90; 90; 87; 65; 22) (e.g., elastic transformation) are applied to regions of the assumed ID sample with higher saliency values, ensuring robustness against style shifts. Given the crafted OOD set and ID set, we apply light augmentation to each set while maintaining the labels to provide various style shifts to each set.

To effectively leverage information from the created sets and develop a robust ND pipeline, we introduce a task-based knowledge distillation strategy (30). Specifically, we use a pre-trained encoder concatenated with a trainable binary classification layer as the teacher and a model trained from scratch as the student. We train the teacher to classify the created ID and OOD sets while only updating the binary layer. Then, using a novel objective function, we force the student to align its output with the teacher when the input is an ID sample and to diverge from the teacher when the input is an OOD sample. The discrepancy between the student and teacher outputs will be utilized as the OOD score at inference time. Our approach is inspired by knowledge distillation, which has proven effective for ND tasks compared to other strategies (19; 78; 23; 32; 93; 11). Notably, our method achieves superior performance compared to both previous knowledge distillation-based and other ND methods, underscoring the effectiveness of our pipeline.

**Contributions:** In this study, we propose a novel data-centric approach along with a new pipeline to achieve a robust and meta-data free ND method. Our strategy, by providing augmented samples obtained through applying style shifts while retaining labels, achieves a more robust representation of distribution shifts. Moreover, through intervening ID samples by identifying and distorting their core regions, we reach synthesized OOD samples. Such samples are then leveraged to make our model more sensitive to the core features. From a causal viewpoint (Refer to Section 4), by sample intervention, as mentioned above, the unwanted correlation between style features and labels is weakened. We note that the general strategy of some previous work (27; 59; 90; 97) that apply hard augmentations on the *entire* image to generate OOD samples, do not necessarily weaken the mentioned unwanted spurious correlation. In addition, our augmentation strategy facilitates the generation of OOD samples whose distribution is potentially closer to that of the real OODs. As well as providing theoretical support to our claims, We evaluate our method on real-world datasets such as autonomous driving and large medical imaging datasets, as well as common datasets such as Waterbird. For comparison, we considered representative and recent ND methods. Our pipeline demonstrates superior results, improving robust and standard performance by up to 12.7% and 6.7% in terms of AUROC, respectively. We further verify our method through a comprehensive ablation study on its different components.

## 2 PROBLEM STATEMENT

**Preliminaries.** The task of ND involves developing a model $f$ to distinguish between two disjoint distributions: ID and OOD. During training, the model only has access to unlabeled ID samples. At inference time, the detector $f$ evaluates a test set, defined as $\mathcal{D}^{\text{test}} = \{\mathcal{D}^{\text{test}}_{\text{ID}} \cup \mathcal{D}^{\text{test}}_{\text{OOD}}\}$, and assesses each test input sample $X$ to determine whether it belongs to ID or OOD by assigning an OOD score $S(X; f)$. Samples exceeding a predefined OOD threshold are classified as OOD. Traditionally, $\mathcal{D}^{\text{train}}$ and $\mathcal{D}^{\text{test}}$ are presumed to originate from identical environments without any style shifts—a prevalent assumption in earlier studies (67; 77). Contrary to this, real-world scenarios often exhibit test samples that diverge in style from the training set. These are represented by $\mathcal{D}'^{\text{test}} = \{\mathcal{D}'^{\text{test}}_{\text{ID}} \cup \mathcal{D}'^{\text{test}}_{\text{OOD}}\}$. Both $\mathcal{D}^{\text{test}}_{\text{ID}}$ and $\mathcal{D}'^{\text{test}}_{\text{ID}}$ retain identical core features, denoted as $X_C$, but vary in style elements, denoted as

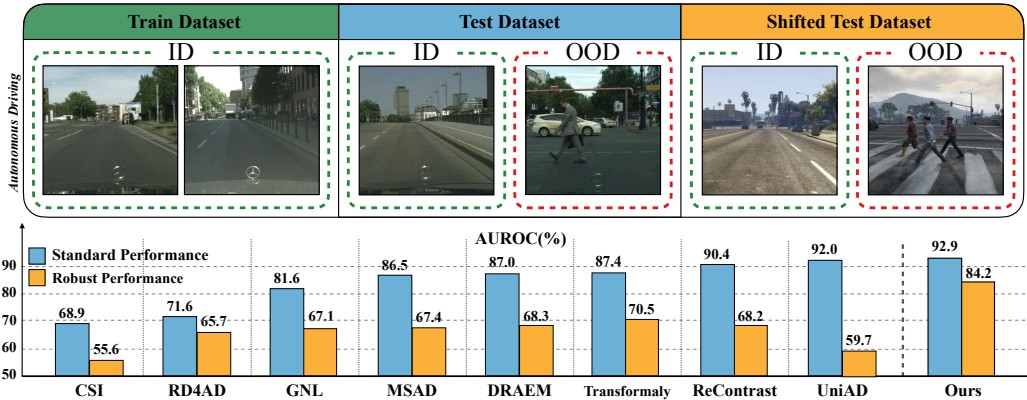

Figure 1: **Evaluating Robust Novelty Detection Performance:** A Comparative Study on the Cityscapes and GTA5 datasets, which both have similar core features but exhibit different style features. Each method has been trained on ID samples from the Cityscapes training dataset, and its performance has been reported on the test sets of Cityscapes (Blue bar) and GTA5 (Orange bar). This highlights the superior performance of our method in contrast to existing methods, which suffer from considerable performance drops. Comprehensive results are provided in Table 1.

$X_E$. Consequently, a robust ND model $f$ should effectively learn and utilize $X_C$ for OOD scoring, while disregarding the style features $X_E$. These concepts are often categorized as informativeness and invariantness, respectively. Using an ideal discriminator $f$, core features can be formally formulated as $\mathcal{S}(X; f) = \mathcal{S}(X_C; f)$, and the relationship between core features and input is expressed through the formula $I(X_C; X) = I(X_C; f(X))$, where $I(\cdot; \cdot)$ denotes the mutual information between the two variables (13; 11; 85; 86; 20).

**Style Bias in Model Training.** In our experiments the training set consisted solely of samples from $\mathcal{D}$ to ensure a fair comparison, as previous methods have mostly developed their pipelines for such scenarios (i.e., designed for a single-dataset setup). However, to avoid a consistent correlation of specific styles with core features (82), we crafted a training set composed of ID samples from both $\mathcal{D}$ and $\mathcal{D}'$, with $\mathcal{D}$ being the dominant source (49; 75; 54; 83). Details of this experiment, including various other ratios, are provided in Appendix A.

For any given ND method, we refer to its detection performance on $\mathcal{D}^{\text{test}}$ as the **standard performance** and on $\mathcal{D}'^{\text{test}}$ as the **robust performance**. It is important to note that we do not have access to metadata that identifies which training samples belong to $\mathcal{D}'$. Additionally, we conduct supplementary experiments using other ratios, specifically 95:5, 90:10, and 80:20, which are detailed in Appendix A. The ratio of 100:0, used in our main results, represents a scenario where no samples from $\mathcal{D}'_{\text{ID}}$ are included in the training data.

## 3 RELATED WORK

**Previous Works on Robust ND.** Recent studies have proposed ND methods for improving robustness under style shifts, including efforts by GNL (11), RedPanda (20), PCIR (13), Stylist (86), and Env-AD (85). These methods, inspired by invariance-inducing approaches such as IRM (2), assume that ID samples are drawn from multiple environments with known styles. Their effectiveness is contingent upon accurately labeled styles in the training data, which can be a significant limitation in datasets where such labels are mostly unavailable or hard to define. Recently, GLAD (98) has shown impressive results on industrial datasets, but still faces severe performance drop on real-world ones. Moreover, GNL proposes to craft different styles by applying minor shifts to ID samples. However, GNL and other models still suffer from performance drops in real-world datasets, as shown in Table 1, which is extensively considered in this study. Importantly, all mentioned methods lack information about potential OOD samples during training, leading to their models struggling with effectively learning core features.

**Transfer Learning for ND.** Several studies (70), including MSAD (69) and UniAD (100), have proposed using ImageNet pre-trained networks. These networks could be useful for ND across

different datasets, such as medical imaging. Among the methods explored, the teacher-student paradigm shows promising results. This approach involves using a pre-trained model as the 'teacher' and a newly trained network from scratch as the 'student'. The main objective is to train the student model while the teacher remains frozen, aiming to mimic the teacher's output on ID samples. The rationale is that the student model, trained exclusively on ID samples, will produce discrepant outputs on OOD samples during the inference phase. Methods such as RD4AD (23), Transformaly (19), and ReContrast (32) are based on this paradigm. More details can be found in Appendix B.

**Auxilary OOD for ND task.** It has been demonstrated that using auxiliary OOD samples during the training step can be beneficial for ND tasks by incorporating an extra dataset (37; 91). Recent works have shown that the effectiveness of this technique largely depends on the diversity and the distance of the distribution of the auxiliary OOD set used during training. In response to this, methods including MIXUP (39), CutPaste (52), and VOS (26) have been proposed. More recently, GOE (48), Dream-OOD (27), and FITYMI (59) address this issue by using large generative models (e.g., Stable Diffusion (73)) for OOD crafting. Interestingly, our crafted auxiliary method does not rely on any extra dataset or generative model. More details about these methods can be found in Appendix B.

## 4 THEORY

**Causal Viewpoint.** From the perspective of causality, the data-generating process can be modeled as the Structural Causal Model (SCM) (66) shown in Fig. 2. In this SCM, $C$ and $E$ denote unobservable causal and non-causal (i.e., domain, environment, or style) variables, from which the observable causal and non-causal components $X_C$ and $X_E$ for an image are obtained. The final image $X$ is the output of $\psi(X_C, X_E)$, where $\psi(.,.)$ is a combining function. The label $Y$ of the image is caused by $X_C$. In the case of spurious correlation, a hidden confounder $U$, would be present such that $E \leftarrow U \rightarrow C$. This creates the path $X_E \leftarrow E \leftarrow U \rightarrow C \rightarrow X_C \rightarrow Y$, which introduces an unwanted correlation between $E$ and $Y$. While there are solutions for when the environment variable $E$ is observable, they are not feasible when domain annotation of samples is not provided. Our approach is effective even in the absence of domain annotation of samples. More precisely, we remove or at least weaken the edge $E \rightarrow X_E$ by intervening on some components of $X_E$ in order to break or loosen the path between $E$ and $Y$, as shown in Fig. 2b. Another orthogonal way of weakening this unwanted correlation is intervening $X_C$ by altering some core features of the ID samples (and correspondingly changing their label to $Y = $ "OOD").

In other words, we want to learn representations that are invariant to changes in $X_E$ and also sensitive to altering $X_C$. By augmenting samples via natural distribution shifts without changing the label, we reduce the correlation of $X_E$ and $Y$. On the other hand, to make our model more sensitive to the causal variables, we synthesize A-OOD samples by altering the core regions of ID images (changing $X_C$ variables and creating samples with $Y = $ "OOD").

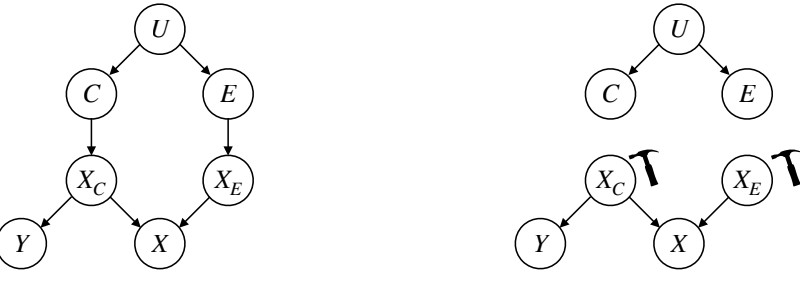

(a) Before applying intervention      (b) After applying intervention

Figure 2: Comparison of causal graphs: Our method, by intervening on $X_E$ and $X_C$, reduces the unwanted spurious correlation between $X_E$ and $Y$. Note that the graph in (b) depicts an *ideal* intervention where full independence between $X_E$ and $Y$ is achieved, which might not fully capture real-world complexities.

**Method Justification.** Although we aim to make the label $Y$ independent of $E$ through the intervention induced by the augmentations, we acknowledge that the achieved independence may be "near-ideal" rather than "ideal". Now we focus on the sufficient conditions that make the intervened $x_C$ "informative," i.e. whether the generated OODs, referred to as A-OODs, are authentically representing the true OODs in their core features.

Let $p_1(x_C)$ and $p_{-1}(x_C)$ represent the distribution of ID and OOD classes on $X_C$, the core features, and $\mathcal{F}$ be the hypothesis space, and for any $f \in \mathcal{F}$, define the expected loss as $Lf := \mathbb{E}_{x_C \sim p}(\ell(f(x_C), y))$, with $p := 0.5p_1 + 0.5p_{-1}$, where $p_1$ and $p_{-1}$ represent the distribution of core sections in ID and OOD classes, respectively. Further, let the expected loss under the A-OOD distribution as $L'f := \mathbb{E}_{x_C \sim p'}(\ell(f(x_C), y))$, with $p' := 0.5p_1 + 0.5p'_{-1}$, where $p'_{-1}$ represents the distribution of A-OOD classes. Further, let $L'_n f$ be the empirical version of $L'f$.

**Theorem 1.** Assume that the input $x$ to the OOD detector lives in a compact space $\mathcal{X}$. The generalization gap in the ID vs. A-OOD learning setup evaluated under real OODs, i.e. $\sup_{f \in \mathcal{F}} |L'_n f - Lf|$, is upper bounded with high probability by the regular generalization bound of learning $f$ in the ID vs. A-OOD learning setup evaluated under A-OOD, added by some factor of the $\ell_2$ distance of real OODs' core distribution $p_{-1}$, and A-OOD core distribution $p'_{-1}$.

*Proof.* Using uniform convergence bounds, one seeks to probabilistically bound $\sup_{f \in \mathcal{F}} |L'_n f - Lf|$. We have:

$$|L'_n f - Lf| = |L'_n f - L'f + L'f - Lf| \leq \underbrace{|L'_n f - L'f|}_{E} + \underbrace{|L'f - Lf|}_{E'}.$$

To bound the difference $E$, one can use the regular generalization bound based on the VC-dimension (92):

$$Lf - L'_n f \leq \sqrt{\frac{1}{n}\left[\left(D\log\left(\frac{2n}{D}\right) + 1\right) - \log\left(\frac{\delta}{4}\right)\right]}$$

with probability of at least $1 - \delta$, where $D$ is the VC-dimension of the $\mathcal{F}$, and $n$ is the training set size. For $\sup_{f \in \mathcal{F}} E'$, we have:

$$E' = \left|\int \ell(f(x_C), y)(p'(x_C) - p(x_C))dx_C\right|$$

$$\leq \underbrace{\sqrt{\int \ell(f(x_C), y)^2 dx_C}}_{E'_1} \underbrace{\sqrt{\int (p'(x_C) - p(x_C))^2 dx}}_{E'_2}.$$

Note that given a compact input space $\mathcal{X}$, both $E'_1$ and $E'_2$ would be bounded. Specifically, considering the fact that $p_1$ is shared between $p$ and $p'$, $E'_2$ corresponds to how much A-OOD and real OOD distributions are close to each other. In addition, $E'_2$ is multiplied by $E'_1$, which is the uniformly weighted average of loss throughout the feature space, which is bounded given a bounded loss function and a compact space $\mathcal{X}$.

**Remarks**: Theorem 1 suggests that once we have an ideal intervention, and the label only depends on $x_C$, it suffices for the intervention to satisfy $p(x_C|do(x_C), do(x_E), Y = \text{"ID"}) \approx p(x_C|Y = \text{"OOD"})$, i.e. the generated OODs through intervention on the *ID* samples $(p(x_C|do(x_C), do(x_E), Y = \text{"ID"}))$ are close in distribution to the real OODs $p(x_C|Y = \text{"OOD"})$. We note that the hard augmentations are minimal alterations on $x_C$ that are needed to turn ID data into OOD. Hence we would expect this specific intervention to make the two mentioned distributions close provided that the real OODs are close to the ID samples. This condition is usually satisfied in real-world OOD detection datasets, where the OOD constitutes minor alterations of the ID samples, which is also known as near-OOD.

## 5 METHOD

**Motivation.** We propose a task-based knowledge distillation method with a novel contrastive-based loss function (16; 35; 42), where the defined task is the classification of ID and crafted OOD samples.

The teacher model aims to update its knowledge by completing the defined task while concurrently encouraging the student model to mimic its behavior closely for ID samples and diverge for OOD samples. To generate informative OOD samples, we propose a simple yet effective method that relies on estimating core regions and distorting them with hard transformations. In the following subsections, we will explain each component of our method, detailing its functionality and benefits.

**Generating Style-Related OOD Samples.** Style-related OOD samples, also referred to as near OOD samples in this study, are those that share stylistic similarities with ID samples but do not belong to the ID set due to differences in core features (90; 59). To generate these style-related OOD samples, we propose a guided strategy that transforms ID samples into OOD by altering the core regions of the ID samples, which contain the primary semantics, while leaving the other regions unchanged.

At first, we define two families of transformations denoted as $\mathcal{T}^+$ (light transformations) and $\mathcal{T}^-$ (hard transformations). $\mathcal{T}^+$ are those that have been shown to preserve semantics in ongoing literature on self-supervised learning (16; 35; 31; 12; 17), while $\mathcal{T}^-$ has been shown to be harmful to preserving semantics in previous studies (39; 1; 29; 90; 87; 65; 22; 45; 52; 84; 44; 61; 104; 14; 25; 101). For crafting OOD samples, we leverage Grad-CAM (81), which provides a saliency map for an input sample using a common pre-trained model (e.g., ResNet18 (34)). Formally, for an ID sample $x$, we randomly choose a light transformation $\tau_1^+ \sim \mathcal{T}^+$. We then compute the saliency map for both $x$ and $\tau_1^+(x)$ and take their element-wise product to ensure the exploited saliency map is style-agnostic. We denote the normalized exploited saliency map as $SM_x$, where higher values correspond to the core features of the assumed ID sample.

For the distortion step, we randomly sample two transformation of harsh transformations $\tau_1^-, \tau_2^- \sim \mathcal{T}^-$. The rationale behind choosing two transformations is to ensure that the distortion shifts the ID sample to OOD. Specifically, for an image $x$ with area $A_x$ and exploited saliency map $SM_x$, we design a mask $m$ that covers an area $\alpha A_x$. We set $\alpha$ randomly between $[0.20, 0.50]$ for each sample to increase the diversity of crafted OOD samples. The mask is then slid over the saliency map, and for each region, the region's weight is determined by summing the pixel values from $SM_x$. Subsequently, we choose $x_{\text{ID}}^{\text{masked}}$ as the core region to distort based on these computed scores. The OOD sample is then created as follows: $x_{\text{OOD}} = \tau_1^-(\tau_2^-(x_{\text{ID}}^{\text{masked}})) + (1 - m) \odot x_{\text{ID}}$. We denote our proposed OOD crafting strategy as $G(\cdot)$, where $x_{\text{OOD}} = G(x_{\text{ID}})$. More details about our generation strategy, including hard transformations and masking approach, can be found in Appendix C. Moreover, samples of the crafted OOD data are presented in Figures 10 and 11. Notably, we conduct extensive ablation studies on various hyperparameters, including $\alpha$ and the backbones, in Appendix D and E. Moreover, the motivation behind using local distortions instead of global ones is more thoroughly discussed in Appendix F. Examples on generated samples are included in Appendix G.

**Task-based Teacher-Student Framework.** Teacher-student (T-S) methods have demonstrated promising results by training a student model to mimic the outputs of a teacher on ID images, using the discrepancy between their outputs as the OOD score (19; 78; 23; 32; 93; 11; 7). However, T-S-based methods experience significant performance drops under style shift scenarios. In our study, we distinguish our approach by proposing a task-based T-S method that considers not only ID but also OOD information to emphasize discriminative features (i.e., core features) during the training step. Moreover, in contrast to previous T-S works that are limited to using frozen teachers, we propose enhancing teacher knowledge by updating its binary-layer's weights.

Formally, we denote the extractors for the student and teacher as $F_s$ and $F_t$, respectively. We extend both extractors by adding a binary layer denoted as $H_s$ and $H_t$. We represent the features extracted by the bottom $l$ layer groups of the teacher model as $F_t^l(\mathbf{x}) \in \mathbb{R}^{w_l \times h_l \times d_l}$, where $w_l$, $h_l$, and $d_l$ denote the width, height, and channel number of the feature map, respectively. We then define the output of the teacher, $f_t(\mathbf{x})$ as follows:

$$f_t^l(\mathbf{x})_k = \frac{1}{h_l \cdot w_l} \sum_{i=1}^{h_l} \sum_{j=1}^{w_l} F_t^l(\mathbf{x})_{jik}, \quad f_t^l(\mathbf{x}) = \frac{f_t^l(\mathbf{x})}{\|f_t^l(\mathbf{x})\|}, \quad f_t(\mathbf{x}) = f_t^1(\mathbf{x}) \oplus \cdots \oplus f_t^l(\mathbf{x}) \oplus H_t(\mathbf{x}),$$

The output of the student, $f_s(\mathbf{x})$, is defined in a similar manner. To reduce computational costs, we transform the 3D features to 1D features by average pooling across channels. This is followed by concatenating the features to form a single vector $f_t(\mathbf{x}) \in \mathbb{R}^{d_l}$ for each sample, which we will use to train the student. We chose $l = 3$, following previous T-S works (32).

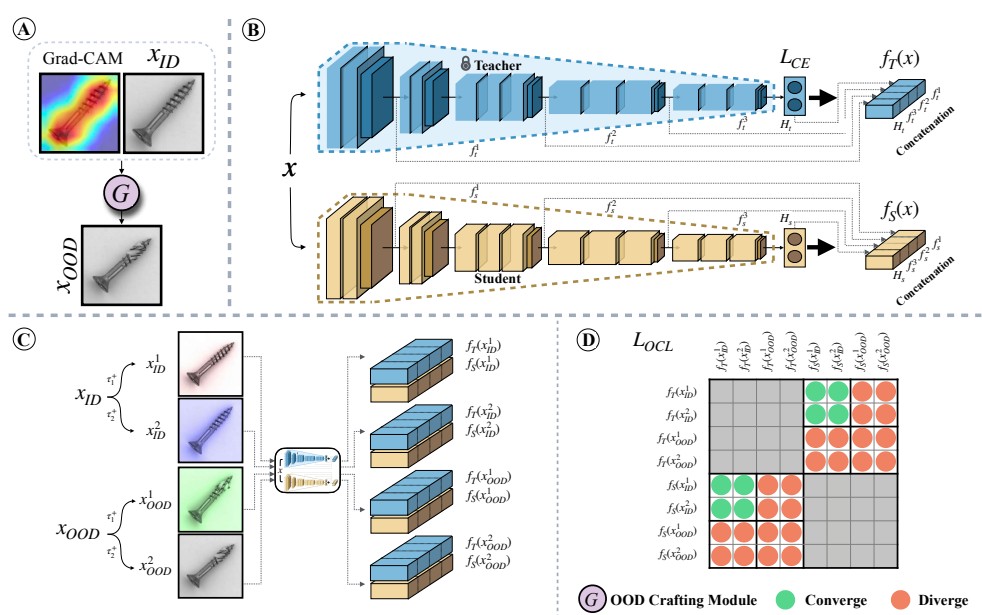

Figure 3: **Overview of our framework for robust novelty detection:** (A) Generation of an auxiliary OOD set by distorting core features of ID samples. (B) Architecture of the proposed pipeline featuring a pre-trained encoder (teacher) and a from-scratch encoder (student), both concatenated to a linear layer. (C) Training step aims to align the output of the student $f_s(\cdot)$ closely with the teacher's output $f_t(\cdot)$ for $x_{\text{ID}}^1$ and $x_{\text{ID}}^2$, and to differentiate them for $x_{\text{OOD}}^1$ and $x_{\text{OOD}}^2$. (D) Green circles indicate pairs where the student's output is intended to be close to the teacher's output, red circles indicate pairs that are meant to diverge, and gray squares represent pairs that have been omitted from the loss function.

**Training Step.** Previous T-S works aimed to define $\mathcal{L}_{\text{TS}}$, which was generally associated with increasing $\text{sim}(f_s(x), f_t(x))$, where $x$ belongs to the ID set. In contrast, we propose an OOD-aware contrastive-based loss, denoted as $\mathcal{L}_{\text{OCL}}$. Specifically, considering a batch of ID training samples, $\mathcal{B}_{\text{ID}} = \{x_i\}_{i=1}^n$, we define $\mathcal{B}_{\text{A-OOD}} = \{x_i\}_{i=n+1}^{2n}$ and $\mathcal{B} = \mathcal{B}_{\text{ID}} \cup \mathcal{B}_{\text{A-OOD}}$, where $\mathcal{B}_{\text{A-OOD}}$ is created using our proposed crafting strategy, i.e., $\mathcal{B}_{\text{A-OOD}} = G(\mathcal{B}_{\text{ID}})$.

For a sample $x$, using $\tau_1, \tau_2 \sim \mathcal{T}^+$, we define $x^1 = \tau_1(x)$ and $x^2 = \tau_2(x)$, and define them as positive pairs, i.e., $P(x^1) = x^2$ and $P(x^2) = x^1$. Then, for each ID sample in $\mathcal{B}$ we define $\mathcal{L}_{\text{OCL}}(x) = \mathcal{L}_{\text{OCL}}(x; f_s, f_t) + \mathcal{L}_{\text{OCL}}(x; f_t, f_s)$, which only updates the student's weights, and $\mathcal{L}_{\text{OCL}}(x; f_s, f_t)$ is defined as:

$$-\sum_{i=1}^{2} \log \frac{\exp(\text{sim}(f_s(x^i), f_t(x^i))/\gamma) + \exp(\text{sim}(f_s(x^i), f_t(P(x^i)))/\gamma)}{\sum_{x' \in \{\tau_1(\mathcal{B}) \cup \tau_2(\mathcal{B})\}} \exp(\text{sim}(f_s(x^i), f_t(x'))/\gamma) + \exp(\text{sim}(f_s(G(x^i)), f_t(x'))/\gamma)} \quad (1)$$

Here, $\gamma$ is the temperature parameter, $\text{sim}(\cdot)$ denotes cosine, and $G(\cdot)$ maps each ID sample to its OOD counterpart, with $G(x_i) = x_{n+i}$ for $1 \leq i \leq n$.

Meanwhile, the teacher is updated using the classification task with cross-entropy loss $\mathcal{L}_{\text{CE}}(\tau_1(\mathcal{B}) \cup \tau_2(\mathcal{B}))$, which is defined on ID and augmented OOD samples. It trains its binary layer while keeping the weights of the other layers frozen. The final loss function for training is $\mathcal{L}_{\text{OCL}} + \mathcal{L}_{\text{CE}}$. A visualization of our method is provided in Fig 3. During test time, we utilize the discrepancy between the teacher and student model as the OOD score, where their features exhibit low differences for ID test samples and high differences for OOD samples due to the defined loss function. Notably, we conduct an ablation study on different options of loss in Appendix H.

# 6 EXPERIMENTS

We validate the efficacy of our proposed robust ND method under style shifts. We conducted an extensive evaluation using a diverse range of industrial and medical datasets, incorporating both natural and synthetic shifts. As shown in Table 1, we compare our method with state-of-the-art ND

methods under both standard and shifted conditions, demonstrating its superior performance across different scenarios. Detailed results of our method, including std over multiple runs, are reported in Appendix I.

**Experimental Setup & Datasets.** To model the distribution shift and conduct evaluation, we followed the setup mentioned in Section 2 for each experiment. We used two datasets, $\mathcal{D}$ and $\mathcal{D}'$, where both include ID and OOD samples. The core features for $\mathcal{D}$ and $\mathcal{D}'$ are the same but come from different environments (different style features). For instance, in the waterbirds experiment, we consider land birds as ID and water birds as OOD. Specifically, we used 3,420 land birds with a land background and 180 land birds with a water background as training data. In the standard test, both land birds and water birds with a land background are considered, while for the shifted test, both land birds and water birds with a water background are used. For the MVTecAD (6) and Visa (110) experiments, similar to GNL, $\mathcal{D}'$ was created manually by us, ensuring that the core features remained constant. For the other experiments, $\mathcal{D}$ and $\mathcal{D}'$ were obtained from existing datasets. Details on $\mathcal{D}$ and $\mathcal{D}'$ for each experiment can be found in Table 2 and Appendix J.

The results in the Table 1 explain each dataset in detail, while the results with $\mathcal{D}$ and $\mathcal{D}'$ swapped are reported in Appendix K. For further details regarding the benchmarks, see Appendix J. Furthermore, extra ablation studies can be found in Appendix A. The Pseudocode for our proposed method is provided in Appendix L. Other evaluation metrics are reported in Appendix M.

**Analyzing Results.** Our approach enhances the average robust detection performance by **12.7%** compared to existing methods (presented in Table 1). Additionally, we achieve a significant improvement of **6.7%** in standard performance. Our evaluation includes methods such as GNL, which was specifically proposed to improve robustness under style shifts, and DRAEM, which uses extra OOD dataset. The results on various challenging datasets demonstrate the applicability of our method in real-world scenarios, all without relying on any metadata or extra dataset. This significant improvement underscores the real-world applicability and generalization of our method.

**Implementation Details.** We utilize a pre-trained ResNet-18 (34) as the foundational encoder network for both the student and teacher networks. Our model undergoes 200 epochs of training using the AdamW (56) optimizer, with a weight decay of 1e-5 and a learning rate of 1e-4. The batch size ($\beta$) for training is set to 128. Further experimental details and time complexity can be found in Appendix N, and limitations of this experimental setup are discussed in Appendix O.

Table 1: Performance of several AD methods, including our proposed method, on multiple pairs of different styles. The results are presented in the format 'Standard/Robust', measured by AUROC (%). 'Standard' represents the scenario where the test set has a similar style to the dominant style in the ID training data, while 'Robust' refers to the scenario where a shifted test set is used, having the same core features but differing in style. Best method on each dataset in terms of Robust performance is highlighted with a blue background.

| Dataset Pair | | Method | | | | | | | | | | |
|---|---|---|---|---|---|---|---|---|---|---|---|---|
| | | CSI | MSAD | DRAEM | RD4AD | UniAD | GLAD | ReContrast | Transformaly | GNL | RedPanda* | Ours |
| *Real-world Datasets* | Autonomous Driving | 68.9 / 55.6 | 86.5 / 67.4 | 87.0 / 68.3 | 71.6 / 65.7 | 92.0 / 59.7 | 89.7 / / 70.1 | 90.4 / 68.2 | 87.4 / 70.5 | 81.6 / 67.1 | 72.8 / 67.3 | 92.9 / 84.2 |
| | Camelyon17 | 60.2 / 53.4 | 70.1 / 64.2 | 68.3 / 59.9 | 60.0 / 56.3 | 62.1 / 56.7 | 70.5 / 62.9 | 59.8 / 60.4 | 64.0 / 63.8 | 65.3 / 60.7 | 68.0 / 65.9 | 75.0 / 72.4 |
| | Brain Tumor | 86.4 / 65.1 | 98.0 / 66.3 | 71.8 / 50.3 | 98.6 / 43.7 | 86.7 / 74.2 | 90.8 / 68.4 | 96.1 / 55.7 | 93.7 / 54.7 | 98.1 / 48.7 | 92.6 / 58.3 | 98.2 / 79.0 |
| | Chest CT-Scan | 59.7 / 54.2 | 70.2 / 58.7 | 67.3 / 66.0 | 64.8 / 59.7 | 70.3 / 60.1 | 65.9 / 61.9 | 66.9 / 60.2 | 71.2 / 70.3 | 63.8 / 58.2 | 67.8 / 60.4 | 72.8 / 71.6 |
| | W. Blood Cells | 62.3 / 45.7 | 76.8 / 60.6 | 67.1 / 60.4 | 61.2 / 53.2 | 55.7 / 60.8 | 64.9 / 59.5 | 59.6 / 50.7 | 79.1 / 57.2 | 60.7 / 56.7 | 74.9 / 56.2 | 88.8 / 72.1 |
| | Skin Disease | 77.2 / 49.5 | 72.1 / 60.3 | 80.4 / 67.2 | 85.1 / 61.9 | 78.9 / 72.5 | 90.0 / 65.7 | 90.5 / 67.3 | 75.4 / 50.1 | 88.3 / 54.8 | 71.7 / 53.9 | 90.7 / 70.8 |
| | Blind Detection | 83.9 / 55.3 | 92.2 / 59.4 | 90.7 / 60.5 | 92.4 / 58.7 | 92.4 / 59.6 | 91.8 / 58.7 | 97.6 / 62.8 | 89.2 / 63.0 | 92.5 / 55.1 | 82.5 / 58.5 | 96.1 / 73.2 |
| *Synthetic Datasets* | MVTec AD | 63.8 / 51.2 | 84.3 / 55.1 | 98.1 / 62.7 | 98.5 / 56.8 | 86.6 / 72.8 | 99.3 / 63.7 | 98.0 / 48.2 | 85.9 / 51.4 | 96.5 / 54.0 | 76.5 / 59.0 | 94.2 / 87.6 |
| | VisA | 65.2 / 53.5 | 84.1 / 63.1 | 96.3 / 58.0 | 96.0 / 64.7 | 84.0 / 70.1 | 99.5 / 60.4 | 91.1 / 54.5 | 85.5 / 53.8 | 89.3 / 60.2 | 84.2 / 65.1 | 89.3 / 82.1 |
| | WaterBirds | 66.8 / 62.3 | 69.2 / 60.4 | 53.1 / 52.5 | 55.9 / 53.6 | 77.1 / 75.0 | 71.8 / 63.7 | 59.4 / 55.3 | 81.0 / 79.3 | 57.1 / 53.9 | 76.8 / 72.4 | 76.5 / 74.0 |
| | DiagViB-MNIST | 89.8 / 72.3 | 84.9 / 58.5 | 83.9 / 63.9 | 77.0 / 53.3 | 63.7 / 55.2 | 83.2 / 59.1 | 76.6 / 54.5 | 67.1 / 55.0 | 65.9 / 65.0 | 83.1 / 76.8 | 93.1 / 73.8 |
| | DiagViB-FMNIST | 87.4 / 74.5 | 90.8 / 55.0 | 87.4 / 67.1 | 78.2 / 64.0 | 74.8 / 50.3 | 80.9 / 60.9 | 77.9 / 60.7 | 84.6 / 63.4 | 75.5 / 64.1 | 85.2 / 71.0 | 92.1 / 78.7 |
| | *Average* | 72.6 / 57.7 | 81.6 / 60.8 | 79.3 / 61.4 | 78.3 / 57.6 | 77.1 / 63.9 | 83.2 / 62.9 | 80.3 / 58.2 | 80.3 / 61.1 | 77.9 / 58.2 | 78.0 / 63.7 | 88.3 / 76.6 |

*Since RedPanda requires metadata for training, we specifically grant access to environment labels for evaluating this method.

Table 2: Specifications of main ($\mathcal{D}$) and shifted $\mathcal{D}'$ pairs for real-world datasets

| Description | Autonomous Driving | Camelyon17 | Brain Tumor | Chest CT-Scan | WBC | Skin Disease | Blind Det. |
|---|---|---|---|---|---|---|---|
| $D$ | Cityscapes (21) | Hospitals 1-3 (10) | Br35H (33) | RSNA (88) | Low Res (107) | ISIC 2018 (18) | APTOS (46) |
| $D'$ | GTA5 (71) | Hospitals 4-5 (10) | Brats 2020 (8) | PD-Chest (47) | High res (107) | PAD-UFES (64) | DDR (53) |

Table 3: An ablation study on our method with the exclusion of different components while keeping the others intact.

| Setups | Components | | | | | Datasets | | | | |
|---|---|---|---|---|---|---|---|---|---|---|
| | A-OOD | Core Estimation | $\mathcal{L}_{CE}$ | $\mathcal{L}_{OCL}$ | $\mathcal{L}_{TS}$ | MVTecAD | Autonomous Driving | MNIST | Waterbirds | Brain Tumor |
| Setup A | - | - | - | - | ✓ | 89.6 / 54.3 | 81.2 / 65.4 | 73.8 / 68.2 | 58.4 / 56.7 | 91.6 / 54.2 |
| Setup B | ✓ | ✓ | - | ✓ | - | 90.3 / 76.9 | 83.1 / 75.3 | 88.0 / 69.7 | 68.3 / 66.1 | 94.1 / 75.7 |
| Setup C | ✓ | ✓ | ✓ | - | ✓ | 91.4 / 72.5 | 84.5 / 78.0 | 85.6 / 69.4 | 75.6 / 67.6 | 91.5 / 63.5 |
| Setup D | ✓ | - | ✓ | ✓ | - | 92.9 / 78.0 | 85.7 / 81.7 | 88.2 / 65.9 | 66.6 / 64.5 | 93.0 / 74.8 |
| Setup E (Ours) | ✓ | ✓ | ✓ | ✓ | - | 94.2 / 87.6 | 92.9 / 84.2 | 93.1 / 73.8 | 76.5 / 74.0 | 98.2 / 79.0 |

## 7 ABLATION STUDY

**Pipeline Components.** To verify the impact of the proposed elements, we conduct comprehensive ablation studies using various datasets. The results are reported in Table 3. In each scenario, we replace certain components with alternative ones while keeping the remaining elements fixed. *Setup A* refers to a scenario where we ignore using auxiliary OOD samples for training and drop the binary classification layers. Instead, we augment ID samples with light transformations and use the common teacher-student based loss function, $\mathcal{L}_{TS}$, for training. Notably, this scenario is similar to the GNL method. *Setup B* highlights the effect of the defined classification task by modifying the training process. Specifically, it excludes the classification task that updates the binary layer of the teacher model. Both the teacher and student models are trained without binary layers. Instead, we train the student model with $\mathcal{L}_{OCL}$ using the created ID and OOD sets. In *Setup C*, we replaced our defined $\mathcal{L}_{OCL}$ with $\mathcal{L}_{TS}$. This tests the efficacy of our proposed loss function in our framework. *Setup D* specifically targets our OOD crafting strategy. Rather than estimating core regions of an ID sample for manipulation, this setup randomly distorts regions of ID samples. This OOD crafting approach is similar to the CutPaste (52) method in terms of finding the region of modification. Results show that *Setup E*, which refers to our proposed (default) framework, achieves superior performance compared to other setups.

**OOD crafting strategy.** In this ablation study, we substituted our OOD crafting strategy with alternative strategies, while keeping other components unchanged. The results, presented in Table 4, demonstrate that our efficient crafting strategy—which does not require an additional dataset or generative model—outperforms other methods. This superiority is based on the fact that other strategies, including MIXUP (39), FITYMI (59), Dream-OOD (27), and GOE (48), fail to preserve the relationship between the style features of created OOD samples and ID samples. Moreover, these methods tend to generate OOD samples biased towards the datasets their backbones are trained on (e.g., Dream-OOD's bias towards LAION (79)), resulting in the creation of distant and unrelated OOD samples (see Figure 10). VOS (26), crafting OOD samples in the embedding space, is ineffective in preserving image style features. CutPaste (52), despite being better than other alternatives, distorts random regions and may alter background features instead of core regions. More details on these methods are in Appendix Q with their implementation details in Appendix R. Experiments on global distortions, rather than local ones, are available in Appendix F.

## 8 CONCLUSION

In this paper, we presented a robust novelty detection method that handles style shifts without requiring metadata. By crafting an auxiliary OOD set and using a task-based knowledge distillation strategy, our approach focuses on core features, reducing the impact of style variations. Evaluations on real-world and benchmark datasets demonstrated significant performance improvements, achieving up to 12.7% higher AUROC compared to existing methods. Our method proves effective in diverse scenarios, offering a robust solution for ND tasks.

Table 4: An ablation study on our method's performance using different A-OOD generation methods.

| OOD Crafting Method | Dataset | | | | | | | Average |
|---|---|---|---|---|---|---|---|---|
| | MVTec AD | Autonomous Driving | MNIST | Waterbirds | Brain Tumor | FMNIST | VisA | |
| MIXUP* | 69.8 / 57.2 | 84.5 / 61.7 | 76.1 / 62.6 | 68.5 / 57.1 | 85.6 / 53.9 | 84.9 / 73.8 | 71.3 / 66.4 | 77.2 / 61.8 |
| CutPaste | 91.7 / 75.1 | 83.6 / 74.8 | 88.2 / 61.9 | 71.9 / 67.0 | 93.8 / 69.3 | 87.8 / 62.6 | 81.9 / 73.2 | 85.6 / 69.1 |
| VOS | 64.2 / 53.9 | 74.8 / 56.1 | 81.3 / 64.0 | 54.8 / 52.3 | 71.8 / 44.2 | 75.4 / 66.2 | 65.1 / 54.8 | 69.6 / 55.9 |
| FITYMI* | 74.0 / 64.5 | 81.6 / 58.4 | 86.9 / 65.8 | 64.5 / 60.9 | 92.7 / 67.4 | 85.1 / 64.7 | 74.6 / 68.2 | 79.9 / 64.3 |
| Dream-OOD* | 86.4 / 75.8 | 87.4 / 76.2 | 84.5 / 56.7 | 82.4 / 71.6 | 79.2 / 63.0 | 82.5 / 61.3 | 69.0 / 57.4 | 81.6 / 66.0 |
| GOE* | 86.8 / 72.7 | 90.5 / 78.3 | 86.1 / 59.2 | 78.3 / 65.2 | 84.1 / 69.7 | 82.1 / 70.6 | 72.8 / 65.7 | 83.0 / 68.8 |
| Ours | 94.2 / 87.6 | 92.9 / 84.2 | 93.1 / 73.8 | 76.5 / 74.0 | 98.2 / 79.0 | 92.1 / 78.7 | 89.3 / 82.1 | 90.9 / 79.9 |

*In contrast to our strategy, these methods employ additional datasets or generative models for crafting OOD data.

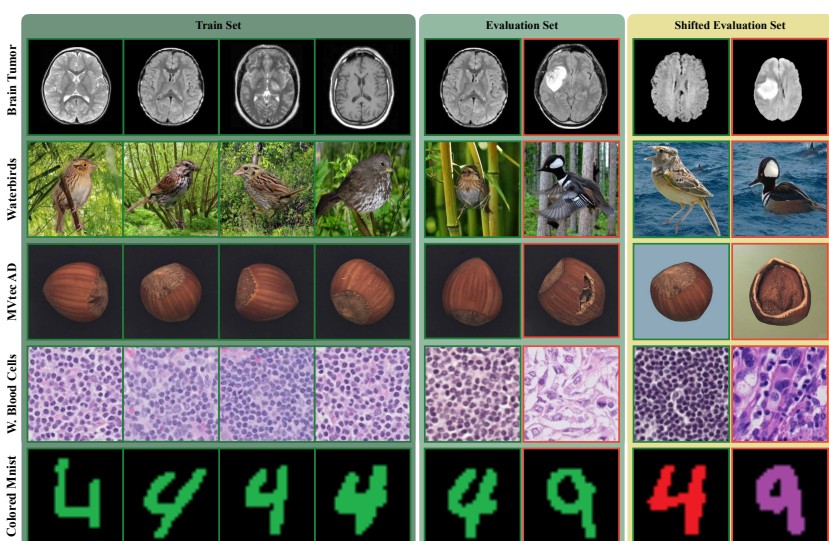

Figure 4: **Examples of Datasets Used in the Study:** This figure illustrates the concept of *Style Shift* in data. We have selected the Brain Tumor Dataset, Waterbirds, MVTecAD, Camelyon17 and Colored MNIST, which perfectly highlight our point. In each row, the left section illustrates 4 images corresponding to the training set of the main dataset, i.e., $\mathcal{D}^{\text{train}}$. The middle section corresponds to the test set of the same dataset, i.e., $\mathcal{D}^{\text{test}}$. The right section corresponds to the samples from the dataset containing style shift, i.e., $\mathcal{D}'$. In the test datasets (middle and right sections), the OOD samples contain a red frame, only for the sake of readability in the figure. Please note that these frames are not available in the actual data. In the brain tumor datasets, images containing a tumor are labeled as OOD and healthy brains are labeled ID, as shown in the figure. The brain images from the main dataset, all include their skulls, which represents itself as a curve around the brain. On the other hand, the images from the shifted dataset do not possess skulls (which could have been removed as a preprocessing procedure). This can lead to the model mistakenly learning the skull as an ID feature, thus labeling all images from the shifted dataset as OOD. In the second row, we consider the waterbirds dataset, which is fully explained in Appendix J.1. In this row, land birds represent ID data and water birds correspond to OOD. In the main dataset (the 2 leftmost columns), the background of all images is a land scenery. In the shifted dataset, all images possess a water background (e.g., sea, lake, etc.). The goal here is to train a model that is robust to the background shifts, and labels images with respect to their foreground, i.e., the type of the bird. In the third row, we consider hazelnut class of the MVTecAD dataset. In this class, non-broken hazelnuts are considered ID, and broken ones are OOD. For the shifted dataset, following the procedure explained for generating synthetic shifted pairs in Appendix J.3, we apply light augmentations on the background of the image, thus simulating a shift in the style, where the style feature here is the background color. Finally, we have the Camelyon17 dataset, which is a lymph node section dataset explained in Appendix J.2. In this set, the ID class represents healthy patients, and the OOD class represents patients with cancerous cells. The shifted dataset has the exact same settings, but the images are taken in a different center, thus facing minor shifts due to difference in equipment, angle, etc. The shift can be seen in the figure as slight changes in the color for both ID and OOD groups, i.e., the shifted images generally have a darker color complex. Note that the Colored MNIST dataset is displayed for intuition only.

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

# APPENDIX

## A EXTRA ABLATION STUDY

### CORRELATION STRENGTH

In this section, we provide results in Table 5 for different amounts of exposure from the shifted training set $\mathcal{D}'^{\text{train}}_{\text{ID}}$ into the main training dataset, examining various correlation strengths. We denote this measure by *correlation strength*. In our default setup, *correlation strength* is set to 0%, indicating the trainset is solely comprised of $\mathcal{D}^{\text{train}}_{\text{ID}}$.

Table 5: An ablation study on the amount of data from $\mathcal{D}'$ which is visible to our model in training time.

| Method | correlation strength = %5(95:5) | | | correlation strength = %10 | | | correlation strength = %20 | | |
|---|---|---|---|---|---|---|---|---|---|
| | MVTec AD | VISA | Autonomous Driving | MVTec AD | VISA | Autonomous Driving | MVTec AD | VISA | Autonomous Driving |
| MSAD | 87.2 / 53.2 | 84.1 / 57.9 | 85.5 / 59.1 | 71.0 / 56.1 | 81.6 / 69.2 | 85.1 / 71.3 | 73.1 / 62.4 | 81.3 / 73.1 | 86.1 / 75.1 |
| Transformaly | 88.5 / 50.6 | 85.5 / 51.3 | 89.1 / 62.4 | 83.2 / 59.4 | 82.7 / 60.1 | 85.4 / 69.1 | 84.1 / 67.0 | 83.5 / 66.5 | 84.3 / 70.3 |
| ReContrast | 99.5 / 50.1 | 97.5 / 50.2 | 90.9 / 58.4 | 96.3 / 55.3 | 89.6 / 63.0 | 88.6 / 72.2 | 95.8 / 60.2 | 86.4 / 68.7 | 88.0 / 74.1 |
| GNL | 98.0 / 52.7 | 90.3 / 58.1 | 84.3 / 65.2 | 96.7 / 58.1 | 87.9 / 65.7 | 80.6 / 71.1 | 94.1 / 65.7 | 86.6 / 69.1 | 81.0 / 73.1 |
| Ours | 95.5 / 86.1 | 90.1 / 81.6 | 93.0 / 79.8 | 93.7 / 89.0 | 88.8 / 84.4 | 91.5 / 86.7 | 93.9 / 91.2 | 89.1 / 86.4 | 90.9 / 89.3 |

## B ADDITIONAL RELATED WORK

**Previous Works on Robust ND**

**Teacher-student based methods for ND.** Efforts to adapt the teacher-student paradigm for ND tasks have involved using a pre-trained model as the teacher and a from-scratch network as the student. The main objective is to train the student model to mimic the teacher's features on ID samples, with the rationale that the student model, trained exclusively on OOD-free samples, will generate discrepant features on OOD samples in inference phase (90). US ensembles several models trained on IDs at different scales to capture a broader spectrum of ID behavior, enhancing the detection of OOD data. Multiresolution Knowledge Distillation (MKD) (78) proposes using multi-level feature alignment to fine-tune the sensitivity to discrepancies between ID and OOD samples. RD4AD (23) advances these methods by using a teacher-student setup with the teacher as an encoder and the student as a decoder focused on feature reconstruction, enhancing detection capabilities. ReContrast (32) introduces a global paradigm for reconstructing teacher features by the student, rather than a regional approach. It also incorporates a stop-gradient operation to stabilize the optimization process.

**Auxiliary OOD Sample Crafting.** CSI (90) and CutePaste (52) propose using fixed hard augmentation to create auxiliary samples. Specifically, CSI relies on Rotation, while CPAD considers CutPaste as a pseudo-OOD. The GOE (48) method employs a pretrained GAN on ImageNet-1K to craft anomalies by targeting low-density areas. FITYM (59) employed an underdeveloped diffusion as a generator. Dream-OOD (27) uses both image and text domains to learn visual representations of normal instances in an embedding space of a pretrained stable diffusion (73) model trained on 5 billion data (e.g. LAION (79)). On the other hand, VOS (26) generates OOD embeddings instead of image data. Notably, we adapt Dream-OOD for generation by using ID sample labels as prompts, as this generative method requires text for generation.

**Previous Robust ND methods.** RED PANDA model propose a robust ND method by focusing on the removal of nuisance attributes by leverageing a domain-supervised disentanglement strategy to learn representations that are invariant to specified nuisance attributes the model shows promise in controlled settings, the effectiveness of RED PANDA is contingent upon the accurate labeling of nuisance attributes in the training data, which can be a significant limitation in datasets where such labels are mostly unavailable or hard to define. calling a method to work without such anotaions. PCIR explores robust Unsupervised ND by aiming to identify invariant causal features across various environments. Specifically, the method assumes that the training data is drawn from multiple known environments, while the test data may come from different, potentially unseen environments. The known environments of each training sample facilitate the development of a regularization term

designed to enhance the model's ability to generalize across diverse environments. Despite the improved robustness demonstrated by their proposed method on specific datasets, a significant limitation is its reliance on the strong presupposition that the environment of each training sample is known. This assumption may not hold in real-world ND scenarios, where datasets often comprise a vast number of samples with unlabelled or unknown environmental contexts.

## C  AUXILIARY OOD GENERATION DETAILS

### MASKING APPROACH

Following our method explanation in Section 5, we wish to find the optimal region of the image to distort. After getting the final normalized saliency map $SM_x$, we use the fact that saliency maps possess spatial coherence, as stated in (81), and look for regions with higher values. The mentioned fact ensures that the selected region's values are continuous, as well as having the core areas covered, resulting in an area of the image that encloses most of the core parts, rather than just including minor and edge areas in it. Noteworthy is that when multiplying the mask by the image, the hard transformation might still get applied to regions with a zero pixel value, i.e., the unmasked area. To tackle this, we crop the region and apply the transformation on the cropped part. Then, we paste the new patch on the original image.

In our primary experiments, the parameter $(\alpha)$, which represents the relative area of the mask with respect to the ID sample, is set between 0.2 and 0.5. This subsection presents an ablation study on various values of $(\alpha)$, with results detailed in Table 6. The findings indicate that variations in $(\alpha)$ have minimal impact on the outcomes, demonstrating that our model is relatively insensitive to changes in this parameter.

Table 6: Exploring the Influence of Random Mask Sizes in Our Method Across Diverse Datasets: A Comprehensive Ablation Study

| Mask Size (% of image) | Dataset | | | | | |
|---|---|---|---|---|---|---|
| | Brain Tumor | Autonomous Driving | DiagViB-MNIST | WaterBirds | MVTec AD | VISA |
| 5% to 20% | 96.2 / 76.1 | 90.0 / 82.1 | 93.0 / 74.2 | 77.0 / 72.3 | 92.1 / 86.7 | 87.8 / 83.0 |
| 10% to 30% | 97.1 / 78.9 | 93.0 / 84.3 | 92.5 / 73.2 | 75.0 / 73.9 | 95.1 / 86.4 | 90.1 / 81.5 |
| 20% to 40% | 98.3 / 79.4 | 91.3 / 83.8 | 93.4 / 72.1 | 75.4 / 73.1 | 94.3 / 85.1 | 89.7 / 81.2 |
| 20% to 50% (Ours) | 98.2 / 79.0 | 92.9 / 84.2 | 93.1 / 73.8 | 76.5 / 74.0 | 94.2 / 87.6 | 89.3 / 82.1 |
| 30% to 50% | 96.9 / 77.6 | 91.7 / 83.1 | 91.3 / 73.0 | 76.1 / 73.6 | 92.8 / 86.6 | 87.1/ 81.5 |
| 40% to 70% | 90.4 / 71.3 | 84.5 / 77.0 | 85.7 / 64.9 | 69.9 / 65.8 | 86.3 / 78.7 | 81.2 / 74.7 |

### AUGMENTATION DETAILS

We apply two types of augmentations to each input $x \in \mathcal{D}^{\text{train}}$, two of which are positive augmentations and two are negative augmentations. The intuition behind this is that with positive augmentations, we seek to make the model understand that light augmentations, which simulate environmental change in actual data, are not decisive in the final decision of the label. Meanwhile, with negative augmentations, we seek to destroy the core of the image, resulting in a new image with different core properties, representing OOD data. We also apply light augmentations to the newly crafted OOD data, to make the model understand environmental changes to this data should not be decisive in the final decision, the same as with ID data.

The exact details on transformations $\mathcal{T}^+$ and $\mathcal{T}^-$ are provided in the main text. For each data, we sample a hard transformation $\tau^+ \in \mathcal{T}^+$. We then attempt to find the core of the image using the procedure explained in our method in Section 5. All transformations are applied using official Python libraries of Albumenations (9) and ImageCorruptions (57).

## D  HYPER PARAMETER SELECTION

We avoided using validation sets and hyperparameter searches, believing our chosen hyperparameters are minimal compared to previous works. Below, we outline our hyperparameters and provide the rationale behind each selection.

Here is a list of all the hyperparameters used in our method and their values:

Table 7: Hyperparameters used in our method

| Mask Ratio | Backbone | Optimizer | Learning Rate | Batch Size | Weight Decay |
|------------|----------|-----------|---------------|------------|--------------|
| 0.75 | ResNet18 | AdamW | 0.0001 | 128 | 0.00001 |

Mask: Please refer to Table 8 in Appendix E. Inspired by the inpainting task, we chose a range of ratios for masking and distorting core regions of an image, following a rule of thumb.

Optimizer, learning rate, weight decay: Common values in literature.

Batch size: High values (typically >100) ensure loss effectiveness.

Backbone: We used ResNet18 as a simple backbone to showcase our model's effectiveness even without complicated architectures. Please refer to 8 for an ablation on different choices of backbones.

## E  BACKBONE

The initial results are reported by applying Resnet18(34) as the backbone of our model due to its efficacy and applicability. However, we conducted extensive ablation studies on the backbones for our T-S model as well as the GradCAM(81) backbone, while preserving the rest of the parameters. As the results indicate, our method demonstrates consistent performance using different pretrained architectures as the backbone. Notably, by using larger architectures as the T/S model, our results further improve.

Table 8: Ablation study on different backbones in our Teacher-Student Model. Each backbone is evaluated under 2 widely used pretrain datasets, Imagenet(24) and Places365(108).

| Backbone | Pretrain | Dataset | | | | | | | Avg |
|----------|----------|---------|---|---|---|---|---|---|-----|
| | | Auto Driving | Camelyon | Brain Tumor | Waterbirds | MVTec | VisA | MNIST | |
| MobileNetV2 | imagenet | 91.7/83.4 | 73.7/71.4 | 97.1/78.8 | 76.2/74.6 | 95.0/88.6 | 88.7/81.2 | 93.1/72.5 | 87.9/78.6 |
| | places365 | 90.9/82.8 | 73.6/70.2 | 96.4/77.4 | 75.8/73.1 | 93.7/86.4 | 87.4/81.7 | 92.6/71.9 | 87.2/77.6 |
| Resnet50 | imagenet | 93.1/84.9 | 76.0/73.7 | 99.1/81.9 | 78.1/73.9 | 95.7/90.1 | 89.6/83.5 | 94.7/75.2 | 89.5/80.4 |
| | places365 | 92.8/85.1 | 75.6/71.6 | 97.9/79.5 | 76.0/74.8 | 94.2/89.1 | 89.1/82.8 | 93.9/73.6 | 88.5/79.6 |
| Wide-resnet50-2 | imagenet | 93.9/86.8 | 76.1/73.6 | 98.8/81.1 | 77.6/75.9 | 95.1/87.4 | 89.8/83.4 | 94.0/74.8 | 89.3/80.4 |
| | places365 | 93.7/86.0 | 75.5/73.9 | 98.7/80.6 | 77.1/75.8 | 94.6/88.1 | 90.2/83.1 | 93.4/74.1 | 89.0/80.2 |
| Wide-resnet101-2 | imagenet | 94.1/84.2 | 77.8/75.9 | 98.7/80.4 | 77.4/75.3 | 94.8/87.0 | 90.3/83.7 | 92.9/73.5 | 89.4/80.0 |
| | places365 | 94.5/84.9 | 77.3/76.1 | 96.8/78.2 | 78.0/74.7 | 94.2/86.6 | 89.5/83.4 | 92.0/72.6 | 88.9/79.4 |
| Vit-b-32 | imagenet | 94.2/84.1 | 76.4/72.9 | 98.6/80.8 | 79.3/77.8 | 94.8/87.1 | 89.9/82.3 | 93.6/73.4 | 89.5/79.8 |
| | places365 | 94.0/84.2 | 76.1/72.3 | 98.3/80.6 | 79.4/77.9 | 94.7/86.7 | 89.5/82.2 | 93.5/73.0 | 89.4/79.5 |
| Resnet18 (Ours) | imagenet | 92.9/84.2 | 75.0/72.4 | 98.2/79.0 | 76.5/74.0 | 94.2/87.6 | 89.3/82.1 | 93.1/73.8 | 88.4/79.0 |

Further, to analyze the effect of Grad-CAM on the backbone, which in our case we again used Resnet18 for similar reasons, we fixed all other components of our model, and ablated on the Grad-CAM-based backbone to see how much it affects the core estimation.

As the results suggest, this component's backbone has minimal effect on the core estimation process and the results, thus highlighting our model's robustness to this component.

## F  IMPACT OF GLOBAL VS. CORE REGION AUGMENTATIONS

To examine the impact of global versus local augmentations on mitigating spurious correlations, we conducted an experiment where the core-region estimation and local distortion strategy were replaced

Table 9: Ablation study on different backbones in our Teacher-Student Model. Each backbone is evaluated under 2 widely used pretrain datasets, Imagenet(24) and Places365(108).

| Backbone | Pretrain | Dataset | | | | | | | Avg |
|---|---|---|---|---|---|---|---|---|---|
| | | Auto Driving | Camelyon | Brain Tumor | Waterbirds | MVTec | VisA | MNIST | |
| Squeezenet1-1 | imagenet | 91.7/82.2 | 73.5/71.4 | 97.5/77.6 | 76.0/72.6 | 93.7/86.7 | 88.7/81.3 | 93.0/72.4 | 87.7/77.7 |
| | places365 | 91.5/81.9 | 73.9/70.9 | 97.9/77.4 | 75.9/72.9 | 93.8/85.9 | 88.6/80.6 | 92.8/71.9 | 87.7/77.4 |
| MobileNetV2 | imagenet | 92.0/82.6 | 74.3/71.3 | 97.8/78.0 | 75.8/73.0 | 94.3/87.3 | 88.6/82.0 | 93.9/72.3 | 88.1/78.1 |
| | places365 | 91.9/83.2 | 74.2/71.5 | 98.2/78.3 | 76.1/73.4 | 94.0/86.5 | 88.7/81.7 | 92.8/72.7 | 88.0/78.2 |
| Efficientnet-b0 | imagenet | 93.4/83.5 | 75.8/73.5 | 97.2/79.6 | 75.7/74.1 | 94.7/87.1 | 89.0/83.4 | 93.7/73.6 | 88.5/79.3 |
| | places365 | 93.7/82.1 | 74.8/72.1 | 97.8/78.5 | 75.2/73.8 | 93.9/86.9 | 88.7/82.6 | 93.6/73.2 | 88.2/78.4 |
| Resnet50 | imagenet | 93.0/84.2 | 76.2/73.1 | 98.6/80.0 | 77.1/75.6 | 95.4/86.7 | 89.5/82.3 | 94.5/74.7 | 89.2/79.5 |
| | places365 | 93.1/83.8 | 77.1/73.2 | 98.7/80.3 | 76.8/75.0 | 95.6/88.4 | 89.4/81.2 | 93.4/72.8 | 89.2/79.2 |
| Wide-resnet101-2 | imagenet | 92.8/84.0 | 75.3/71.6 | 98.5/79.4 | 75.6/75.1 | 95.0/88.1 | 90.2/83.8 | 93.8/74.1 | 88.7/79.4 |
| | places365 | 92.6/83.9 | 74.8/73.0 | 97.1/78.9 | 76.8/74.7 | 94.8/86.3 | 89.5/83.0 | 94.7/72.9 | 88.6/79.0 |
| Vit-b-32 | imagenet | 93.2/84.6 | 75.8/71.9 | 97.5/80.7 | 76.5/74.6 | 95.1/89.5 | 91.5/84.8 | 91.6/71.7 | 88.7/79.7 |
| | places365 | 93.1/84.1 | 76.6/72.6 | 96.7/80.2 | 77.9/75.1 | 93.9/88.6 | 90.1/84.5 | 91.4/71.0 | 88.5/79.4 |
| Resnet18 (Ours) | imagenet | 92.9/84.2 | 75.0/72.4 | 98.2/79.0 | 76.5/74.0 | 94.2/87.6 | 89.3/82.1 | 93.1/73.8 | 88.5/79.0 |

with a global hard transformation. This approach evaluates the effectiveness of global augmentations, which apply distortions uniformly across the entire image, compared to localized augmentations that focus on core regions. The results, presented in the table below, indicate that global transformations lead to a decrease in OOD detection performance. This supports our hypothesis that selectively distorting core regions, while preserving style features, is more advantageous for the model's ability to detect distribution shifts and differentiate between ID and OOD samples.

Table 10: Comparison of Global Augmentation and Core Region Augmentation (Ours) on Various Datasets

| Method | Datasets | | | | | | | | | | |
|---|---|---|---|---|---|---|---|---|---|---|---|
| | Driving | Camelyon | Brain | Waterbirds | MVTec | VisA | Chest | Blood | Skin | MNIST | Avg |
| Global Aug | 91.9/81.0 | 74.5/72.1 | 93.3/70.3 | 72.5/74.2 | 89.8/81.5 | 84.0/77.3 | 71.9/64.8 | 80.4/66.5 | 87.1/71.4 | 90.7/70.9 | 83.6/65.9 |
| Core Region Aug (Ours) | 92.9/84.2 | 75.0/72.4 | 98.2/79.0 | 76.5/74.0 | 94.2/87.6 | 89.3/82.1 | 72.8/71.6 | 88.8/72.1 | 90.7/70.8 | 93.1/73.8 | 87.2/76.8 |

# G EXAMPLE OF DATASETS USED IN THE STUDY

In this section, we present examples of both real-world and synthetic datasets, along with their corresponding shifted datasets that demonstrate variations in style features used in this study. For the brain tumor detection task, the *Br35H* dataset is employed, with the shifted dataset being the *Brats 2020* dataset. As for the *Camelyon17* dataset (10), data from hospitals 1-3 constitute the main dataset, while data from hospitals 4-5 serve as the shifted dataset. As for the *Waterbirds* dataset, the main dataset consists of land birds with land backgrounds as the ID set and water birds with land backgrounds as the OOD set. The shifted dataset includes land birds with water backgrounds and water birds with water backgrounds. Examples for these datasets are provided in Figure 4.

For the *MVTecAD* and *VisA* datasets, we apply Meta's Segment Anything Model (SAM) (50) to alter the background of the objects. Additionally, for texture modifications, we center-paste the image onto a random ImageNet dataset sample. Examples for the *MVTecAD* dataset are illustrated in Figure 7, and for the *VisA* dataset, examples can be seen in Figure 8.

# H LOSS FUNCTION ANALYSIS

## DEVELOPMENT PROCESS

The core concept behind using A-OOD in the T-S architecture is to encourage the student model to produce outputs that are closer to the teacher model's outputs when the input is an ID sample, and to diverge further when the input is an OOD sample.

**Setup A:** At first glance, it seems that adding a simple term to the common cosine similarity of the T-S models can help, specifically:

$$= \text{sim}(f_s(x_{\text{ID}}), f_t(x_{\text{ID}})) - \text{sim}(f_s(x_{\text{A-OOD}}), f_t(x_{\text{A-OOD}})) \tag{2}$$

where $f_s$ and $f_t$ are the student and teacher models, respectively, and $x_{\text{ID}}$ and $x_{\text{A-OOD}}$ represent in-distribution and auxilary out-of-distribution samples. However, the results in Table 11 show that this method is not a suitable option for robust novelty detection. Based on this observation and recognizing the effectiveness of contrastive learning in distinguishing between similar and dissimilar samples, we decided to introduce a novel T-S architecture where the student mimics the teacher using a contrastive learning loss instead of cosine similarity.

**Setup B:** The first solution that comes to mind to enhance contrastive learning with A-OOD is the following loss function:

$$
= -\sum_{i=1}^{2} \log \frac{\exp(\text{sim}(f_s(x^i), f_t(x^i))/\gamma) + \exp(\text{sim}(f_s(x^i), f_t(P(x^i)))/\gamma)}{\sum_{x' \in \{\tau_1(\mathcal{B}) \cup \tau_2(\mathcal{B})\}} \exp(\text{sim}(f_s(x^i), f_t(x'))/\gamma)}
$$

$$
+ \sum_{i=1}^{2} \log \frac{\exp(\text{sim}(f_s(G(x^i)), f_t(G(x^i)))/\gamma) + \exp(\text{sim}(f_s(G(x^i)), f_t(P(G(x^i))))/\gamma)}{\sum_{x' \in \{\tau_1(\mathcal{B}) \cup \tau_2(\mathcal{B})\}} \exp(\text{sim}(f_s(G(x^i)), f_t(x'))/\gamma)}, \quad (3)
$$

where $f_s, f_t, P, G$ are the same as defined in Section 5. In this loss, inspired by contrastive loss (90), we try to make the student mimic the outputs of the teacher to ID samples. Simultaneously, the second term tries to make the outputs of the student to the OOD samples close to those of the teacher. Then, the second term is subtracted from the first, indicating that we want their similarity minimized, resulting in their divergance. However, in this scenario, the loss function operates unstably, and the results in Table 11 show that it is not a robust OOD detection model.

**Setup C:** Next, we propose our novel loss function in equation (1), which ensures stable training and enables the student model to produce outputs that are closer to the teacher model's outputs for ID samples, while diverging further for OOD samples.

$$
\mathcal{L}_{\text{OCL}}(x) = \mathcal{L}_{\text{OCL}}(x; f_s, f_t) + \mathcal{L}_{\text{OCL}}(x; f_t, f_s) \tag{4}
$$

**Setup D:** For further exploration and ablation study of our method, we removed $\mathcal{L}_{\text{OCL}}(x; f_t, f_s)$ and observed its effect.

$$
\mathcal{L}_{\text{OCL}}(x) = \mathcal{L}_{\text{OCL}}(x; f_s, f_t) \tag{5}
$$

**Note**: In all setups (A, B, C, D), we also include the $\mathcal{L}_{\text{CE}}$ term in the loss.

Table 11: Performance comparison of different proposed losses. The table shows the evaluation results of different losses, including our proposed loss, highlighting their effectiveness and stability.

| Loss setup | Dataset | | | | | |
|---|---|---|---|---|---|---|
| | Brain Tumor | Autonomous Driving | DiagViB-MNIST | WaterBirds | MVTec AD | VISA |
| Setup A | 88.7 / 62.1 | 83.2 / 68.9 | 82.8 / 58.3 | 63.7 / 60.1 | 79.3 / 65.8 | 81.0 / 67.3 |
| Setup B | 85.4 / 64.0 | 73.6 / 65.1 | 79.8 / 61.7 | 60.3 / 56.4 | 81.7 / 66.0 | 78.6 / 65.3 |
| Setup C (Ours) | 98.2 / 79.0 | 92.9 / 84.2 | 93.1 / 73.8 | 76.5 / 74.0 | 94.2 / 87.6 | 89.3 / 82.1 |
| Setup D | 96.2 / 75.2 | 88.5 / 79.1 | 90.0 / 71.1 | 73.4 / 72.1 | 90.3 / 83.6 | 84.4 / 75.9 |

STABILITY OF LOSS

In the analysis of various configurations applied to the Cityscapes dataset, the distinctions in performance and loss metrics are clearly illustrated (Figures 5a and 5b). Figure 5a displays the AUROC curves for four different setups, where it is evident that our setup, Setup C, not only achieves faster convergence but also delivers comparatively higher AUROC values. Similarly, Figure 5b shows the normalized loss across these setups, with Setup C exhibiting a considerably more consistent loss trajectory than its counterparts. Notably, Setups A and B demonstrate significant fluctuations in their loss metrics, indicating a lack of stability. While Setup D has similar performance to Setup C, the consistency and rapid convergence of Setup C affirms its superiority.

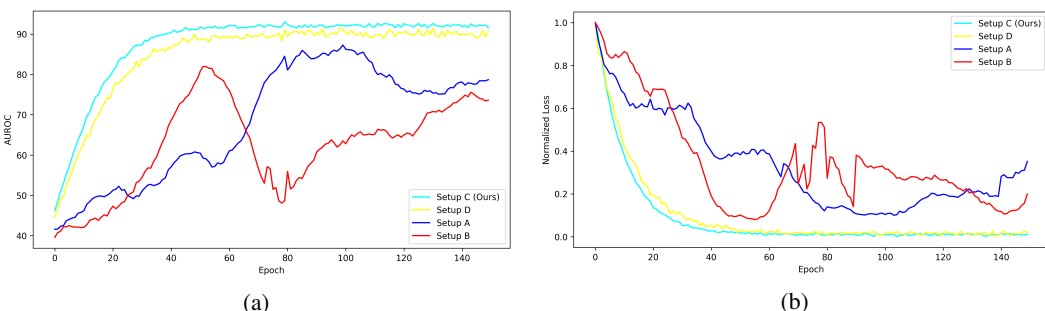

(a)            (b)

Figure 5: Performance and Loss Comparison Across Different Setups on the Cityscapes Dataset: Figure (a) showcases the AUROC curves for four setups, highlighting that Setup C (Ours) not only converges more rapidly but also achieves superior performance relative to the others. Figure (b) presents the normalized loss, where Setup C demonstrates a notably stable loss profile. In contrast, Setups A and B display less stability, with fluctuations in their loss metrics. These comparisons underscore the efficiency and robustness of our approach in both performance and stability.

## I DETAILED RESULTS

In this section, in Tables 12 and 13, we provide the mean and standard deviation of our method's results on the provided datasets in Table 1 using 5 different seeds. These were not reported in the main table due to space constraints.

Table 12: Detailed results of our method's performance on the first 6 datasets, over 5 runs.

| Method | Dataset | | | | | |
|---|---|---|---|---|---|---|
| | Autonomous Driving | Camelyon | Brain Tumor | Chest CT-Scan | White Blood Cells | Skin Disease |
| Ours ($\mathcal{D}$) | $92.9 \pm 0.51$ | $75.0 \pm 0.64$ | $98.2 \pm 0.12$ | $72.8 \pm 0.68$ | $88.8 \pm 0.61$ | $90.7 \pm 0.43$ |
| Ours ($\mathcal{D}'$) | $84.9 \pm 0.62$ | $72.4 \pm 0.84$ | $79.0 \pm 0.20$ | $71.6 \pm 0.83$ | $72.1 \pm 0.75$ | $70.8 \pm 0.52$ |

Table 13: Detailed results of our method's performance on the second 6 datasets, over 5 runs.

| Method | Dataset | | | | | |
|---|---|---|---|---|---|---|
| | Blind Detection | MVTecAD | VisA | Watebirds | Diag-MNIST | Diag-FMNIST |
| Ours ($\mathcal{D}$) | $96.1 \pm 0.91$ | $94.2 \pm 1.01$ | $89.3 \pm 0.76$ | $76.5 \pm 0.67$ | $93.1 \pm 0.21$ | $92.1 \pm 0.32$ |
| Ours ($\mathcal{D}'$) | $73.2 \pm 0.98$ | $87.6 \pm 1.21$ | $82.1 \pm 0.89$ | $74.0 \pm 0.75$ | $73.8 \pm 0.34$ | $78.7 \pm 0.28$ |

## J DATASETS

In the following paragraphs, we explain how we obtain $\mathcal{D}^{\text{train}}$, $\mathcal{D}^{\text{test}}$, and $\mathcal{D}'^{\text{test}}$. One detail shared among all datasets is that after obtaining the datasets, we add $k$ samples from the shifted dataset, $\mathcal{D}'_{\text{ID}}$ to the training data, where $k$ is equal to $5\%$ of the size of $\mathcal{D}^{\text{train}}$. Worth noting is that our model significantly outperforms other models, even in the absence of this added data, as explained in Section 6. This detail is not mentioned in the following paragraphs to avoid redundancy.

### J.1 DETAILS ON BENCHMARK DATASETS WITH SYNTHETIC SHIFTS

- **DiagViB-MNIST and DiagViB-FMNIST** (28) we use the DiaViB-6 benchmark dataset for our experiments, DiaViB-6 provide a unique capability to manipulate five key generative factors in colored images: texture overlays, object dimensions, placement, brightness, and saturation, in addition to semantic features corresponding to the label. Adjusting these factors enabled the

creation of diverse environments varying in these six aspects. All images in both datasets were resized to dimensions of $3 \times 256 \times 256$. The main dataset contained data from two environments, while the shifted dataset consisted of data from five distinct, previously unseen environments. In both DiagViB-MNIST and DiagViB-FMNIST datasets, the DiagViB-6 benchmark employed class 4 as the ID set, with class 9 assigned as the OOD set. These datasets are publicly available under the AGPL-3.0 license.

- **WaterBirds** (76) We evaluated our method using the Waterbird dataset, which contains natural images with distribution shifts caused by changes in the background habitat, alternating between aquatic and land settings. In our experiments, the main dataset includes land birds with land backgrounds as the ID set and water birds with land backgrounds as the OOD set (5% of the training data comes from the ID set of the shifted dataset). The shifted dataset includes land birds with water backgrounds and water birds with water backgrounds. The main dataset's training data consists of 3,420 images with land backgrounds and 180 images with water backgrounds. The test set of the main dataset contains 3,551 images with land backgrounds. The shifted dataset, used for evaluation, includes 4,637 images with water backgrounds. All images are resized to $224 \times 224$. This dataset is publicly released under the MIT license.

## J.2 DETAILS ON NATURAL SHIFT DATASETS

- **Autonomous Driving** The main dataset used for Autonomous Driving is Cityscapes (21). This dataset provides stereo videos from 50 cities, with detailed annotations for 30 classes, including roads and buildings. Intuitively, to reflect real-world scenarios, we want the streets with few obstacles (e.g. pedestrians) to be considered "safe", thus being labeled as ID, while the crowded streets be labeled unsafe, i.e. OOD. We utilize Cityscapes by extracting $256 \times 256$ patches from the center of the images to construct an OOD detection dataset. In our methodology, we classify roads, sidewalks, buildings, walls, fences, poles, vegetation, terrain, sky, cars, trucks, and buses as ID classes, while all other classes are treated as OODs. Each patch is labeled as OOD if it contains any object from an OOD class; otherwise, it is labeled as ID. The license clearly states that the dataset is made freely available for both academic and non-academic purposes, and permission to use is given.

  The robust pair of Cityscapes is the GTA5 dataset (71). The GTA5 dataset consists of 24,966 synthetic images with pixel-level semantic annotations, generated using the open-world video game Grand Theft Auto 5. Similarly, we extract $256 \times 256$ patches from the center of these images to form another OOD detection dataset. The ID classes remain the same as in the Cityscapes dataset, whereas the OOD classes include trains, motorcycles, persons, riders, traffic signs, traffic lights, and bicycles. Their code is released under the MIT license.

- **Camelyon17** We use the Camelyon17 dataset (10; 51) which is a lymph node section dataset gathered from patients with potential breast cancer. The images are taken from tissue patches obtained from five different hospitals, each potentially having a tumorous tissue within other parts of the tissue. The ID data is defined as healthy tissues and tumorous tissues are labeled as OOD. We use the train data from the first 3 hospitals (218,510 images) as the training data. We then use the test data from the first 3 hospitals (99,121 images) as the main test data, and the test data from hospitals 4 and 5 (77,862 images) as the shifted test data. All images are resized to $224 \times 224$. This dataset is publicly released under the CC0 1.0 license.

- **Brain Tumor** The main dataset is Br35h (33), which consists of 3,000 magnetic resonance images (MRIs) of human brains, with 1,500 images of tumorous brains and 1,500 of non-tumorous brains. We split the non-tumorous set 70/30, training on 70% of the non-tumorous data and evaluating on the remaining non-tumorous and tumorous images during test time. The shifted pair is the Brain Tumor (8) dataset, which contains 3,764 MRIs of human brains. These images are also categorized into two classes: tumorous and non-tumorous. Similar to the Br35h dataset, we split the non-tumorous set 70/30, training on 70% of the non-tumorous data and evaluating on the remaining non-tumorous and tumorous images during test time. All images are resized to $224 \times 224$. Both datasets are free to public use under the CC BY 4.0 license.

- **Blindness Detection** Blindness Detection is a pair of datasets dedicated to images of color fundus, with the main dataset being APTOS, which is the official training dataset released for the 2019 APTOS blindness detection challenge (46). This dataset contains 3,662 images with grades 0-4 indicating the severity of Diabetic Retinopathy (DR). We used the images with grade 0 (1,805

images) as ID, and the rest as OOD. As for the shifted dataset, we used the DDR dataset (53), which contains 13,673 fundus images from 147 hospitals in China. Similar to APTOS, these images are also classified into 5 groups according to DR severity: none, mild, moderate, severe, and proliferative DR. We label the images with no DR severity (6,266 images) as ID, and the rest as OOD. All images are resized to 224× 224. Both datasets are publicly available under the MIT license.

- **Skin Disease**   Skin Disease is a pair of image datasets dedicated to different skin diseases. The main dataset is ISIC2018, which is the publicly available dataset of the ISIC2018 Lesion Diagnosis challenge (18). It contains seven classes corresponding to seven different categories of skin disease. We take the NV (Nevus) class as ID, and the rest as OOD, following the setup used in (106) and (32). The training set comprises 6,702 ID images. The shifted dataset is PAD-UFES-20 (64), a skin lesion dataset composed of clinical images collected from smartphones. It contains 2,298 total images, with 224 of them labeled NEV (Nevus), which we take as ID, and the rest are taken as OODs. All images are resized to 224×224. The ISIC dataset is available under CC-BY-NC license, and the DDR dataset is under CC-BY-4.0 license.

- **Chest CT-Scan**   Chest CT-Scan is a pair of datasets dedicated to images of frontal view chest X-RAY images. The main dataset, RSNA, which is available from the 2018 RSNA Pneumonia Detection Challenge (88), consists of images of 30,227 patients, with 9,555 of them diagnosed with Pneumonia. The shifted dataset is another pneumonia dataset used for image classification, which is used by Kermany et. al (47). It contains 5,856 images in total, with 1,341 of them being ID and the rest being defected. To create the training dataset, we use 70% of the ID data, and use the rest of them for testing the model. All images are resized to 224×224. RSNA license is available for non-commercial purposes, and the shifted dataset is licensed under CC-BY-4.0.

- **White Blood Cells**   The White Blood Cells (WBC) dataset (107), comprises two sets of datasets, each containing microscopic images of 5 different cell types. In our setup, from each dataset, cells with the label "Lymphocite" are taken as ID and the rest are taken as OOD. The main dataset contains three hundred 120×120 images of WBCs and their color depth is 24 bits. The shifted dataset contains one hundred 300×300 color images with significantly higher resolution. To obtain training data, we sample 70% of the ID images from the main dataset, resulting in 123 images. The rest of dataset 1 are used as the main test data, and dataset 2 is used as the robust test data. All images are resized to 224×224. WBC is under the GPL-3.0 license.

### J.3   DETAILS ON OUR APPROACH TO GENERATING SYNTHETIC SHIFTED PAIRS

The MVTec Anomaly Detection (MVTecAD) dataset (6) is specifically designed for evaluating anomaly detection methods in industrial settings. It features high-resolution images from 15 different categories, including both objects like screws and textures like leather, each with examples of ID and defective conditions. We utilized the MVTecAD dataset as the main dataset in our experiments. For the robust version, we added a 10% width padding to all ID and OOD images in the MVTecAD test set for texture categories. Additionally, for object categories, we modified the background color of the MVTecAD test set using Facebook's SAM (Segment Anything Model)(50) model. MVTecAD is under the CC-BY-NC-SA 4.0 license.

The VisA dataset (111) introduces a novel and substantial dataset, comprising a total of 10,821 images, with 9,621 labeled as ID and 1,200 as OOD, doubling the size of MVTec. This dataset is organized into 12 subsets, which are divided into three standard categories based on object properties. The first category includes four printed circuit boards (PCBs) with intricate structures. The second category consists of datasets showcasing multiple instances in a single view, such as Capsules, Candles, Macaroni1, and Macaroni2. The third category comprises single instances with roughly aligned objects, like Cashew, Chewing gum, Fryum, and Pipe fryum. In our experiments, the main dataset utilized is VisA, and for the robust version, we altered the background color of the VisA test set using Facebook's SAM (Segment Anything Model)(50) model. VisA is under the CC-BY 4.0 license.

## K   INTERCHANGED DATASET PAIRS RESULTS

In this section, we provide results for the case where the "Main" and "Shifted" datasets are interchanged, i.e. $\mathcal{D}$ is used as the Shifted dataset and $\mathcal{D}'$ is the Main dataset. The splitting policies for

train and test datasets, and exposure percents are the same as the original setup. Results are presented in Table 14, and descriptions of the datasets are provided in Table 15.

Table 14: Performance of some AD methods, including our proposed method, on the interchanged pairs of datasets given in Table 15. The results are presented in the format "Standard/Robust", measured by AUROC (%). "Standard" represents the scenario where the test set has a similar style to the dominant style in the ID training data, while "Robust" refers to the scenario where a shifted test set is used, having the same core features but differing in style.

| | Dataset Pair | Method | | | |
|---|---|---|---|---|---|
| | | UniAD | ReContrast | Transformaly | Ours |
| *Real-world* Datasets | Autonomous Driving | 78.6 / 70.5 | 83.7 / 71.9 | 89.1 / 72.3 | 88.3 / 79.3 |
| | Camelyon | 69.7 / 58.4 | 68.7 / 62.1 | 70.9 / 63.7 | 78.9 / 72.1 |
| | Brain Tumor | 90.4 / 63.1 | 88.1 / 67.5 | 81.0 / 68.4 | 90.4 / 80.0 |
| | Chest CT-Scan | 73.6 / 61.7 | 76.2 / 60.7 | 78.4 / 62.3 | 80.0 / 73.8 |
| | W. Blood Cells | 69.8 / 60.7 | 75.1 / 54.7 | 72.1 / 66.7 | 80.1 / 69.3 |
| | Skin Disease | 82.1 / 60.7 | 85.1 / 61.2 | 79.1 / 64.1 | 88.1 / 72.3 |
| | *Average* | 77.3 / 62.5 | 79.4 / 63.0 | 78.4 / 66.3 | 84.3 / 74.5 |

Table 15: Specific $D$ and $D'$ sets for each Real-world dataset

| Description | Autonomous Driving | Camelyon17 | Brain Tumor | Chest CT-Scan | WBC | Skin Disease | Blind Det. |
|---|---|---|---|---|---|---|---|
| $D$ | GTA5 (71) | Hospitals 4-5 (10) | Brats 2020 (8) | PD-Chest (47) | High res (107) | PAD-UFES (64) | DDR (53) |
| $D'$ | Cityscapes (21) | Hospitals 1-3 (10) | Br35H (33) | RSNA (88) | Low Res (107) | ISIC 2018 (18) | APTOS (46) |

## L   ALGORITHM

In this section, we present the Robust Novelty Detection Algorithm that outlines our method, detailed further in Section 5. The `A-OOD-Generator` function is designed to generate an OOD sample from a given ID sample. Meanwhile, the `ViewGenerator` function constructs two positive views for each ID and OOD sample, utilizing a series of random positive augmentations.

During training, the `A-OOD-Generator` function produces $X_{\text{OOD}}$ from $X_{\text{ID}}$, and subsequently, the `ViewGenerator` function generates positive views of both $X_{\text{ID}}$ and $X_{\text{OOD}}$. These views are then fed into the network. The loss is computed according to equation (1), following which the model is updated.

**Algorithm 1** Robust Novelty Detection

**function** A-OOD-GENERATOR($X_{ID}$)

$\tau^+$ = sample({Color Jitter, Horizontal Flip, Grayscale, ...})

$S_{X_{ID}} = Grad(X_{ID}) \odot Grad(\tau^+(X_{ID}))$    ▷ Get saliency map for $X_{ID}$ using Grad-CAM

$mask$ = get_mask($X_{ID}, S_{X_{ID}}$)

$\tau^-$ = sample({Rotation, Elastic, Distortion, ...})    ▷ T is a sample of hard augmentations

$X_{OOD} = mask \odot \tau^-(X_{ID}) + (1 - mask) \odot X_{ID}$

**return** $X_{OOD}$

**end function**

---

**function** VIEWGENERATOR($X_{ID}, X_{OOD}$)

$T_1, T_2, T_3, T_4$ = Sample({Color jitter, Blur, Random H-flip, . . . })

▷ $T_i$s are samples of light augmentations

**return** $T_1(X_{ID}), T_2(X_{ID}), T_3(X_{OOD}), T_4(X_{OOD})$

**end function**

---

**function** TRAIN

**for** $X_{ID} \in Dataloader$ **do**

$X_{OOD}$ = A-OOD-generator($X_{ID}$)

$X = [X_{ID}, X_{OOD}]$

$X_{ID}^{view1}, X_{ID}^{view2}, X_{OOD}^{view1}, X_{OOD}^{view2}$ = ViewGenerator($X_{ID}, X_{OOD}$)

$Y = [0] \times |X_{ID}| + [1] \times |X_{OOD}|$

▷ Y is a label vector where 0 denotes samples from $X_{ID}$ and 1 denotes samples from $X_{OOD}$.

$loss = \mathcal{L}_{OCL}(X_{ID}^{view1}, X_{ID}^{view2}, X_{OOD}^{view1}, X_{OOD}^{view2}) + \mathcal{L}_{CE}(X, Y)$   ▷ As defined in equation (1)

Update($loss$)

**end for**

**end function**

---

**function** MAIN

**for** epoch in range(200) **do**

Train()

**end for**

**end function**

---

$Main()$

## M    EXTRA EVALUATION METRICS

The AUROC (Area Under the Receiver Operating Characteristic curve) metric is a widely recognized metric for evaluating the performance of outlier detection methods. To provide a more comprehensive assessment, we have included results using two additional metrics—AUPR and FPR95%—previously employed in related studies (38). The table below contrasts our method with TRANSFORMALY, a recent outlier detection technique. Specifically, FPR95% measures the false positive rate at which 95% of outlier samples are accurately identified; a lower FPR95% indicates enhanced detection capabilities. Both AUROC and AUPR encapsulate a method's effectiveness across various thresholds, where a higher AUROC suggests a greater probability that an outlier is correctly prioritized higher than an in-distribution sample based on anomaly scores. Therefore, higher values of AUROC and AUPR are indicative of superior performance, with a baseline uninformative detector achieving an AUROC of 50%.

## N    IMPLEMENTATION DETAILS

### MODEL DETAILS

We employ a pre-trained ResNet-18 as the foundational encoder network for both the student and teacher ResNet-18 models, excluding the binary layers from each. To classify ID and auxiliary OOD data, we append a new linear layer at the end of the network. Additionally, we extract features from layers 1, 2, and 3 of both the student and teacher models to calculate the OCL loss. These intermediate features, which provide information at various levels of abstraction, are crucial for the student model to effectively mimic the teacher model.

### TRAINING AND EVALUATION DETAILS

During optimization, our model is trained for 200 epochs using the AdamW optimizer, with a weight decay of 1e-4 and a learning rate of 5e-5. The batch size for training is set to 128. We evaluated all methods using the Area Under the Receiver Operating Characteristic curve (AUROC). Our experiments were conducted on NVIDIA GeForce RTX 3090 GPUs (24GB) using Python v3.8.

### TIME COMPLEXITY

An additional component in our work that adds to the time complexity, in comparison with previous ND works, is the saliency map extraction from GradCAM. Using the resources explained in the previous subsection, we generate saliency maps for one hundred 224×224 images in $\sim 2.7 \pm 0.04$ seconds over all datasets in our setup. Notably, we compute these maps for each sample before starting the training phase. This adds an initial overhead but reduces overall time complexity as we avoid redundant computations of the maps.

Moreover, we observe that our method usually converges after $\sim 150$ epochs on average, which should be taken into consideration when estimating total time. For the batch size and backbone specified in Appendix N, each epoch should take less than one minute. Further, evaluation time is proportional to dataset size, but for an average-sized dataset, e.g. One-class MVTecAD, should be less than a minute. Formally speaking, calculating the $\mathcal{L}_{\text{OCL}}$ loss takes $O(\beta^2)$ time, giving $O(GradCAM) + \text{(total iters)} \cdot (O(\beta^2) + O(\mathcal{L}_{\text{CE}}))$. On eval time, we have $(|\mathcal{D}'^{\text{test}}|) \cdot O(f)$, where $f$ is the output of the model.

## O    LIMITATIONS

In this study, we utilize an interpretable method to identify and distort the core features of ID samples. Despite demonstrating the effectiveness of our approach, there are some limitations to consider. Firstly, in certain image domains, such as texture images (e.g., grid images), the distortions introduced may resemble random alterations rather than systematic ones, potentially impacting the performance of the method because the core regions of texture images are not well defined. Secondly, although our method has been validated on 12 diverse datasets spanning various tasks, including white blood cell analysis in medical imaging, the hard augmentations applied may not always accurately

represent real-world OOD samples. This discrepancy could affect the performance of our approach in specific scenarios where the real-world OOD samples significantly differ from the crafted OOD samples.

Furthermore, our proposed method can be viewed as a general pipeline that consistently performs well, rather than achieving large margins in standard novelty detection on a specific dataset. Although this low-variance performance may indicate a level of reliability, in certain scenarios, a highly specialized method with higher performance might be more desirable, as discussed in Appendix T.

## P  EMBEDDING-LEVEL ANALYSIS

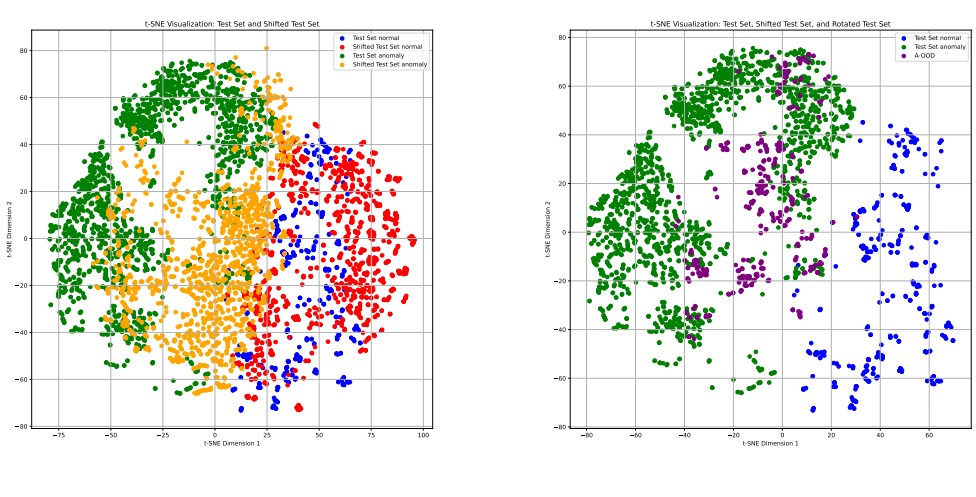

(a) Embeddings of main and shifted dataset      (b) Main test set embeddings against A-OODs

Figure 6

To provide more insight into our model's intrinsic discriminative abilities, we plot the visualizations of the embeddings generated by our model on the BRaTS dataset. Below we give explanations and insights on these plots.

Figure 6a provides a t-SNE visualization demonstrating the model's ability to differentiate between normal and anomalous samples in both "Standard (main test set)" and "Robust (shifted test set)" settings, as achieved by our Student model. In the plot, blue and green dots represent ID and OOD samples in standard settings, and red and yellow dots represent ID and OOD samples in robust settings, respectively. In our pipeline (outlined in Section 5), features from the three layers, $f_s^1(x)$, $f_s^2(x)$, and $f_s^3(x)$, are concatenated along with the binary head. For this visualization, we focus solely on the concatenated features from the three layers, excluding the binary head, to analyze the embedding structure. The plot illustrates that, even prior to the application of the binary classification layer, our model successfully distinguishes between normal and anomalous samples under both standard and robust setups, demonstrating strong performance. This shows the effectiveness of the embedding features in capturing distinctions between the different sample categories. Specifically, in standard settings, the discrimination is handled with near-perfect precision, and in robust settings, we observe a minor performance decline.

Figure 6b illustrates the embeddings generated by our model for the actual test set compared to those for our crafted Auxiliary Out-of-Distribution samples (A-OODs). The purpose of this visualization is to provide a direct comparison between the distribution of the A-OODs and that of the actual OODs. As highlighted in Theory 1 in Section 4, the ideal A-OODs should closely approximate the distribution of the actual OODs, facilitating the model's capacity to generalize effectively. Additionally, as mentioned in the Remarks, our A-OODs are carefully designed by making minimal modifications to in-distribution (ID) samples, ensuring they exhibit out-of-distribution characteristics without diverging excessively. This plot reinforces that assertion by demonstrating that the A-OODs align

closely with the optimal distribution. Consequently, this proximity helps the model train a more robust discriminator, which is better equipped to distinguish between ID and OOD samples.

## Q    OOD GENERATION METHODS COMPARISON

In this section, we present examples of OOD generation methods, including our own A-OOD generation method, detailed in Section 5. The comparative samples can be viewed in Figure 10 for the MVTecAD dataset, in Figure 11 for the VisA dataset, and in Figure 9 for the remaining datasets. Techniques such as *Fake It*, *Mixup*, and *Dream OOD* influence both the core and style features of the samples. In contrast, the *CutPaste* method, which selects pasting areas randomly, may variably affect either core or style features, thus not consistently impacting the sample label. However, our method, as demonstrated in Section 5, specifically targets and distorts the core features of the samples, demonstrating its efficacy in generating OOD samples from given ID samples.

Specifically, for the *Dream OOD* technique, we provided the desired label in the form of text.

## R    DETAILS ON EVALUATING OTHER METHODS

To obtain the results of other models in our experiment, we use the official code released with their work. We train and evaluate their code with minimal changes, i.e. only changing the dataloaders and code related to that. Moreover, for works with multiple setups (e.g. backbone, loss function, etc.) we use the default method reported in their paper. As for epoch number, batch size, and other hyperparameters, we set them to their default values reported in their papers.

## S    CONSISTENT SUPERIOR PERFORMANCE WHEN ENCOUNTERING FAR OOD SAMPLES

While our method, alongside the localized augmentation strategy, has primarily focused on approximating the behavior of near-OOD samples, it is crucial to evaluate robustness against diverse types of OOD data. To this end, we conducted an experiment simulating far OOD samples, as detailed in Table 16.

Table 16: Performance on ID and OOD Datasets

|  | **OOD Dataset** | | | |
| --- | --- | --- | --- | --- |
| **ID Dataset** | Main testset | Shifted testset | Gaussian Noise | SVHN |
| MVTec AD | 94.2 | 87.6 | 97.9 | 100.0 |
| Driving | 92.9 | 84.2 | 100.0 | 98.7 |
| Camelyon | 75.0 | 72.4 | 97.4 | 100.0 |
| Brain | 98.2 | 79.0 | 100.0 | 95.7 |
| VisA | 89.3 | 82.1 | 99.3 | 100.0 |
| WaterBirds | 76.5 | 74.0 | 95.6 | 100.0 |
| Chest | 72.8 | 71.6 | 96.3 | 100.0 |
| Blood | 88.8 | 72.1 | 98.1 | 100.0 |
| Skin | 90.7 | 70.8 | 97.2 | 100.0 |

## T    COMPARISON WITH SOTA METHODS ON MVTECAD AND VISA.

Recent anomaly detection methods have primarily focused on two datasets: MVTecAD and VisA, achieving impressive results on these datasets. However, these methods often struggle to generalize to other datasets. In this section, we compare our method with additional state-of-the-art (SOTA) methods, which incorporate a heavy inductive bias, on the MVTecAD and VisA datasets. The primary objective of our study was to propose a robust model for the novelty detection domain, particularly under distributional shifts. Our method addresses the critical challenge where models tend to rely on non-causal (style) features of the data rather than causal ones. To overcome this, we

Table 17: Performance of our method vs. best previous work on multiple datasets, using the AUPR and FPR95% metrics.

| Method | Metric | Dataset | | | | | | |
| --- | --- | --- | --- | --- | --- | --- | --- | --- |
| | | Brain Tumor | Autonomous Driving | MNIST | FMNIST | WaterBirds | MVTecAD | VISA |
| Ours | AUROC | 98.2 / 79.0 | 92.9 / 84.2 | 93.1 / 73.8 | 92.1 / 78.7 | 76.5 / 74.0 | 94.2 / 87.6 | 89.3 / 82.1 |
| Ours | AUPR | 95.7 / 81.9 | 91.0 / 86.6 | 85.1 / 76.1 | 96.0 / 80.9 | 72.1 / 69.1 | 96.4 / 89.7 | 92.6 / 84.7 |
| Ours | FPR95% | 5.7 / 27.4 | 13.4 / 19.9 | 6.3 / 35.8 | 16.0 / 32.3 | 19.1 / 28.5 | 15.3 / 22.4 | 17.6 / 25.0 |
| Transformaly | AUROC | 93.7 / 54.7 | 87.4 / 70.5 | 67.1 / 55.0 | 84.6 / 63.4 | 81.0 / 79.3 | 85.9 / 51.4 | 85.5 / 53.8 |
| Transformaly | AUPR | 95.1 / 61.9 | 89.1 / 72.9 | 71.0 / 58.5 | 87.1 / 66.7 | 84.1 / 79.9 | 88.1 / 53.8 | 82.6 / 59.8 |
| Transformaly | FPR95% | 10.6 / 48.7 | 17.3 / 33.1 | 31.8 / 45.9 | 25.6 / 36.1 | 15.4 / 26.5 | 16.9 / 37.9 | 16.2 / 43.0 |

| Method | Driving | Camelyon | Brain | Chest | Blood | Skin | Blind | MVTec | VisA | Waterbirds | DiagViB-FMNIST | Avg | Clean.Std. |
| --- | --- | --- | --- | --- | --- | --- | --- | --- | --- | --- | --- | --- | --- |
| SimpleNet (55) | 82.6 / 63.7 | 64.7 / 54.5 | 89.1 / 60.2 | 62.4 / 50.7 | 61.4 / 54.1 | 82.2 / 64.7 | 86.7 / 58.4 | 99.6 / 65.1 | 96.8 / 71.0 | 68.1 / 59.8 | 78.8 / 58.3 | 79.3 / 60.0 | 13.5 |
| DDAD (62) | 86.4 / 65.2 | 65.3 / 59.7 | 90.9 / 61.4 | 60.2 / 45.8 | 60.9 / 52.7 | 84.2 / 65.1 | 91.8 / 57.1 | 99.8 / 62.6 | 98.9 / 60.4 | 64.8 / 58.7 | 76.5 / 61.6 | 80.0 / 59.1 | 15.1 |
| EfficientAD (4) | 86.1 / 70.1 | 68.4 / 59.6 | 91.5 / 65.7 | 61.9 / 52.2 | 63.7 / 54.3 | 86.7 / 63.4 | 88.6 / 60.3 | 99.1 / 59.7 | 98.1 / 57.5 | 65.7 / 59.1 | 78.3 / 59.4 | 80.7 / 60.1 | 13.8 |
| DiffusionAD (102) | 84.8 / 61.9 | 67.6 / 63.4 | 88.7 / 63.3 | 63.0 / 54.8 | 60.2 / 56.1 | 85.7 / 64.0 | 87.3 / 61.7 | 99.7 / 67.1 | 98.8 / 63.8 | 66.8 / 63.1 | 75.8 / 60.7 | 79.9 / 61.8 | 14.0 |
| ReconPatch (41) | 83.9 / 69.3 | 68.0 / 56.9 | 87.6 / 59.6 | 62.8 / 55.1 | 55.9 / 53.7 | 85.7 / 64.0 | 89.7 / 57.6 | 99.6 / 60.2 | 95.4 / 61.2 | 65.0 / 60.5 | 76.8 / 59.3 | 77.2 / 59.7 | 14.9 |
| GLASS (15) | 85.3 / 66.7 | 68.1 / 57.4 | 90.4 / 63.7 | 63.7 / 57.7 | 63.5 / 54.1 | 87.2 / 62.7 | 90.3 / 60.7 | 99.9 / 65.3 | 98.8 / 62.7 | 68.4 / 61.7 | 79.7 / 63.7 | 81.4 / 61.5 | 13.6 |
| GeneralAD (89) | 89.5 / 73.9 | 69.1 / 64.2 | 91.4 / 71.0 | 64.5 / 62.7 | 65.7 / 63.1 | 89.7 / 66.4 | 88.3 / 57.1 | 99.2 / 67.2 | 95.9 / 64.9 | 70.3 / 65.7 | 78.3 / 64.7 | 82.0 / 65.5 | 12.7 |
| GLAD (98) | 89.7 / 70.1 | 70.5 / 62.9 | 90.8 / 68.4 | 65.9 / 61.9 | 64.9 / 59.5 | 90.0 / 65.7 | 91.8 / 58.7 | 99.3 / 63.7 | 99.5 / 60.4 | 71.8 / 63.7 | 80.9 / 60.9 | 83.2 / 63.3 | 12.9 |
| Ours | 92.9 / 84.2 | 75.0 / 72.4 | 98.2 / 79.0 | 72.8 / 71.6 | 88.8 / 72.1 | 90.7 / 70.8 | 96.1 / 73.2 | 94.2 / 87.6 | 89.3 / 82.1 | 76.5 / 74.0 | 92.1 / 78.7 | 87.9 / 76.9 | 8.9 |

Table 18: Comparison of our method with state-of-the-art methods on MVTecAD and VisA, showing superior average performance and lower clean performance variability across both clean and robust conditions (The implementation of these methods is based on their official repositories, utilizing their default hyperparameters.)

introduced a novel pipeline, backed by theoretical analysis. While certain SOTA methods may achieve better performance on specific datasets in clean settings, our method demonstrates superior average performance across both clean and robust (shifted) scenarios. Furthermore, our method exhibits a lower standard deviation in clean performance, highlighting its reliability and consistency. Based on our results, we believe our method strikes the best balance between clean and robust performance compared to previous approaches (Table 18).

Recently the MVTecAD and VISA datasets have attracted attention as they have high relevance for industrial novelty detection benchmarks. Notably though, although they have achieved nearly perfect performance on these challenging datasets, they suffer from over-specialization (60), meaning the algorithms that achieve high performance lack generalization abilities. Thus, we acknowledge that our method's clean performance on the MVTec/VISA dataset could be perceived as a drawback. However, we believe it is not entirely fair to compare our method against standard novelty detection methods on specific datasets like MVTec/VISA, and our clean performance should be considered as a non-critical limitation.

The SOTA methods for these datasets have often been developed with a strong inductive bias, specifically tailored to the characteristics of these datasets. For example, these methods, such as PatchCore with a 99.6% AUROC performance on MVTecAD, heavily rely on patch-based feature extraction. This approach works well because these datasets primarily consist of texture-based novelty samples rather than semantic ones. This reliance can lead to performance degradation when these methods are applied to other novelty detection datasets that emphasize semantic novelty detection. This issue is also reflected in the computed standard deviation, which was lower for our method than for the SOTA methods on MVTecAD/VISA. (An example of texture-based novelty detection: a broken screw versus an intact screw. An example of semantic-based novelty detection: a dog versus a cat, with the cat assumed as an inlier concept.)

When comparing our method to the existing novelty detection approaches, we believe our method demonstrates superior robustness in detection performance, while maintaining clean detection performance that is consistently higher than or competitive with other methods (see Table 1). Additionally, we would like to highlight that our average clean performance across datasets surpasses that of other methods, indicating that our solution offers the best overall trade-off among existing methods.

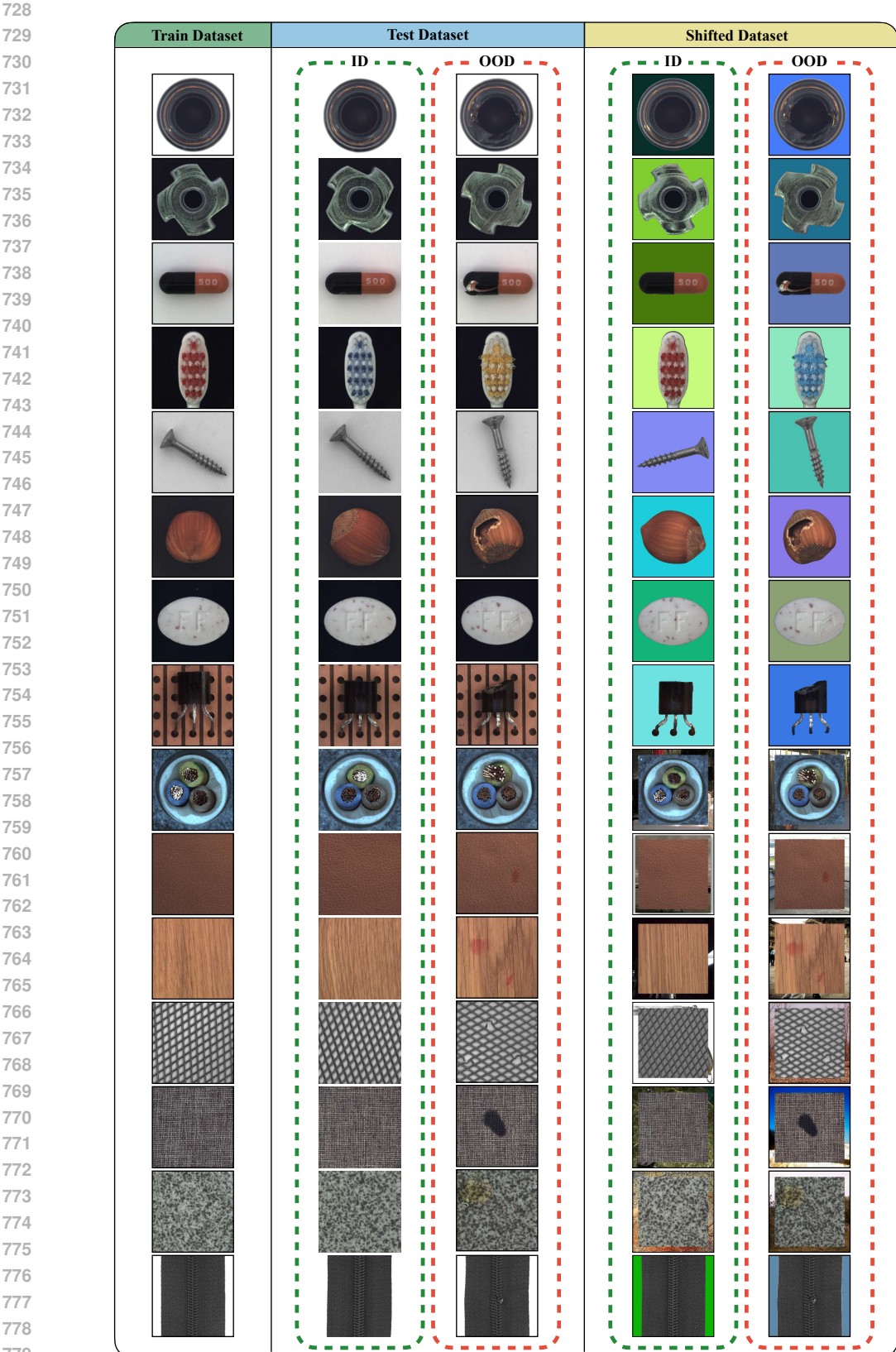

Figure 7: **Main and Shifted datasets comparison on the MVTec AD dataset.**

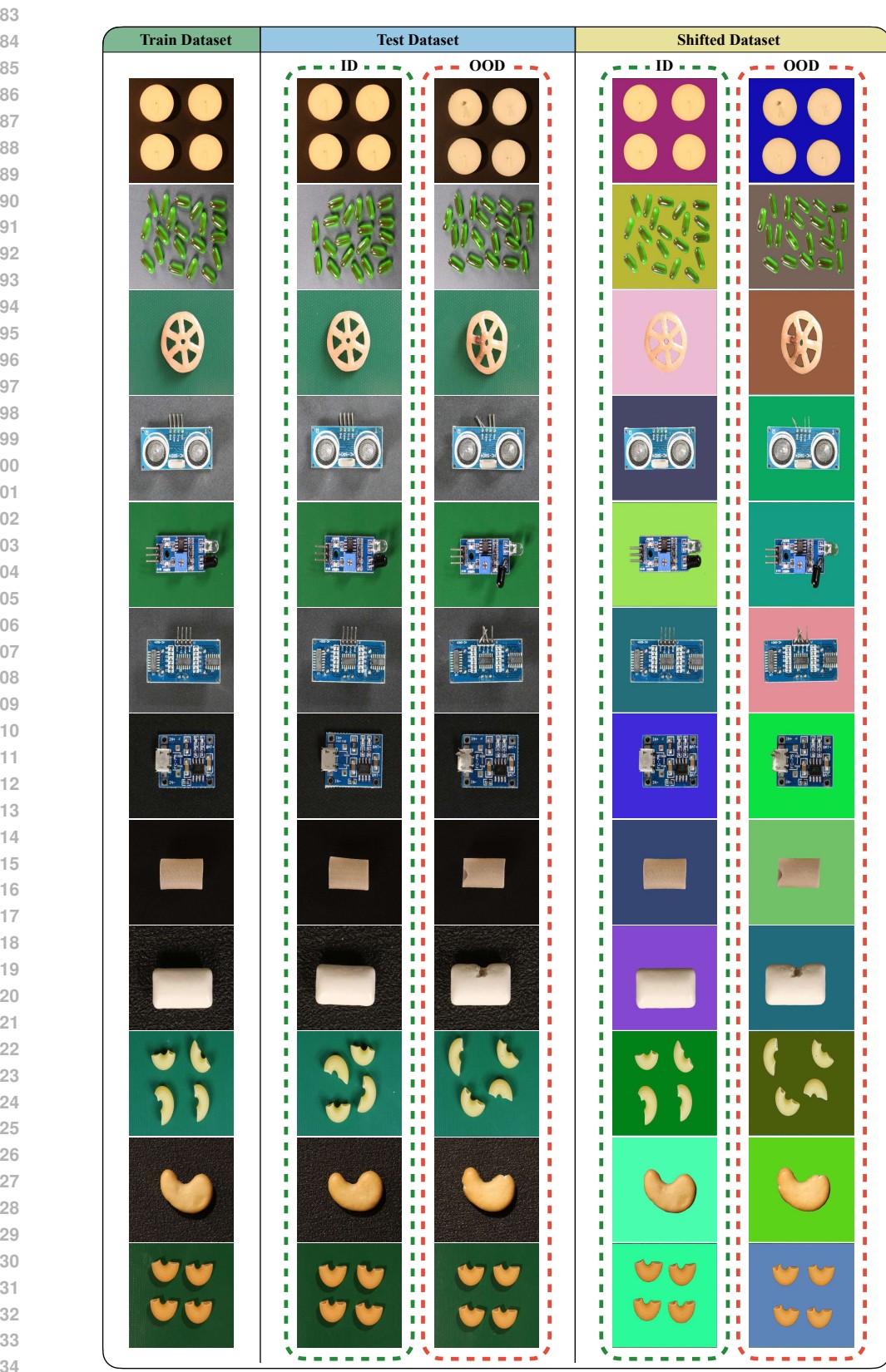

Figure 8: **Main and Shifted datasets comparison on the VisA dataset.**

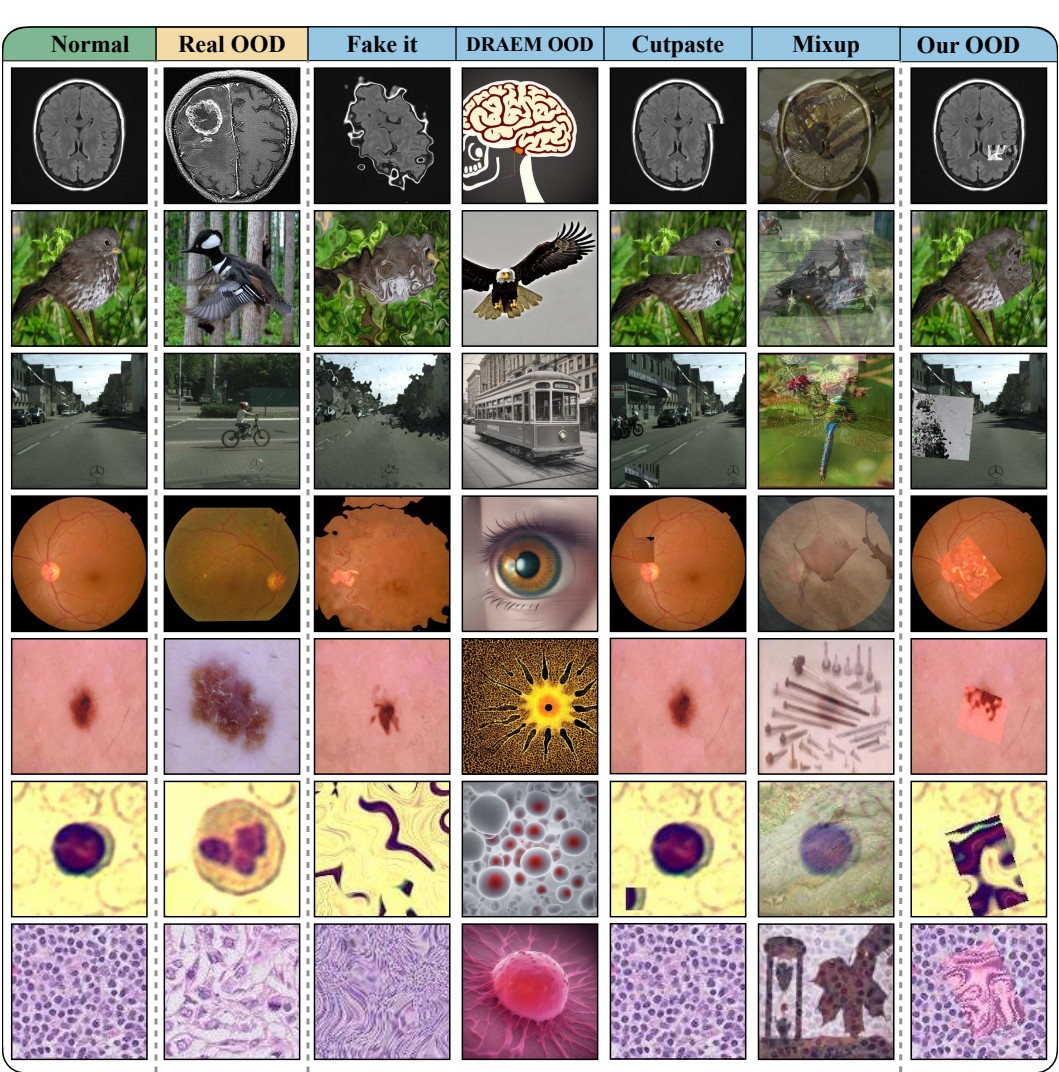

Figure 9: **OOD Generator methods comparison on datasets.**

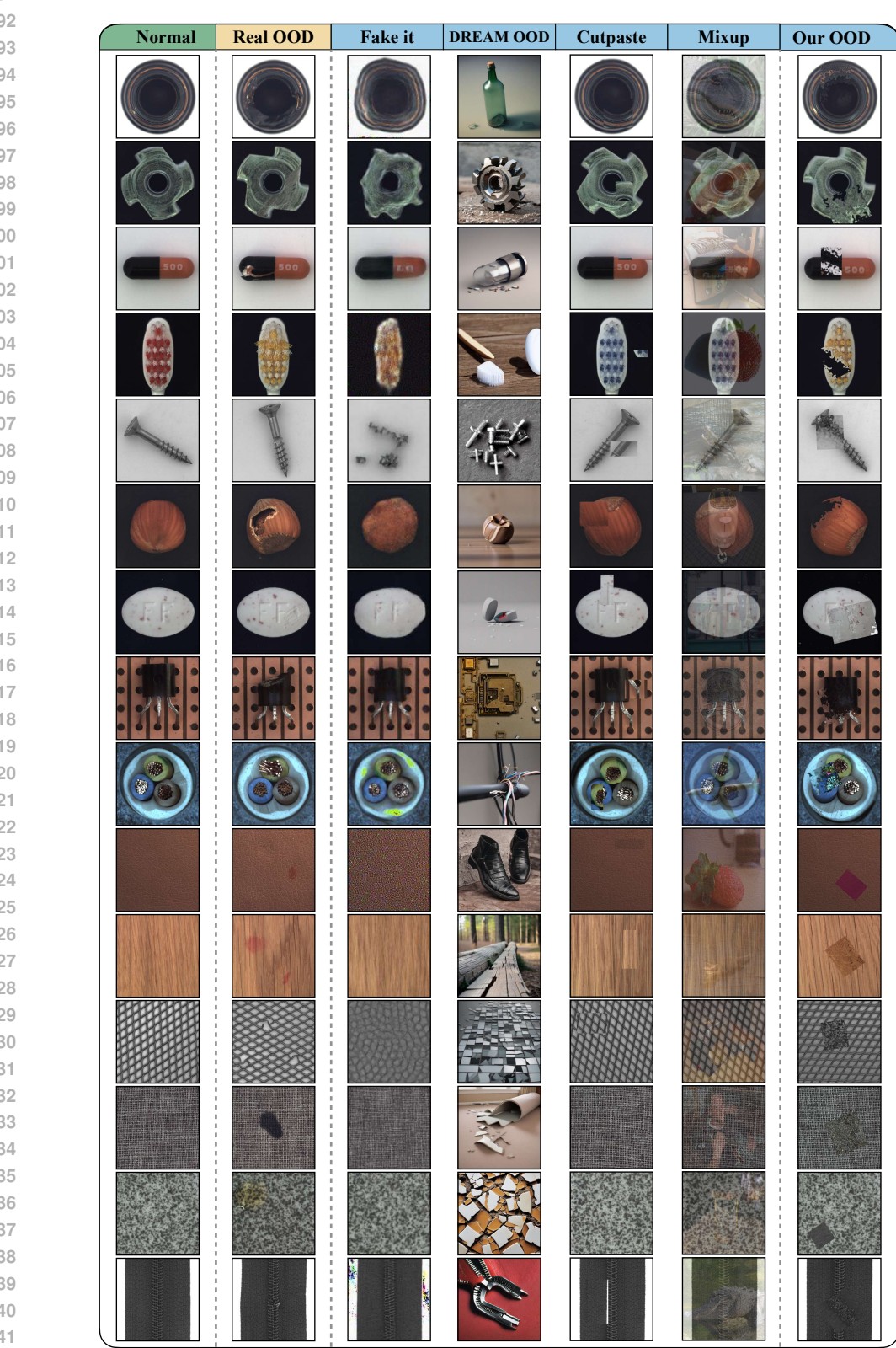

Figure 10: **OOD Generator methods comparison on the MVTec AD dataset.**

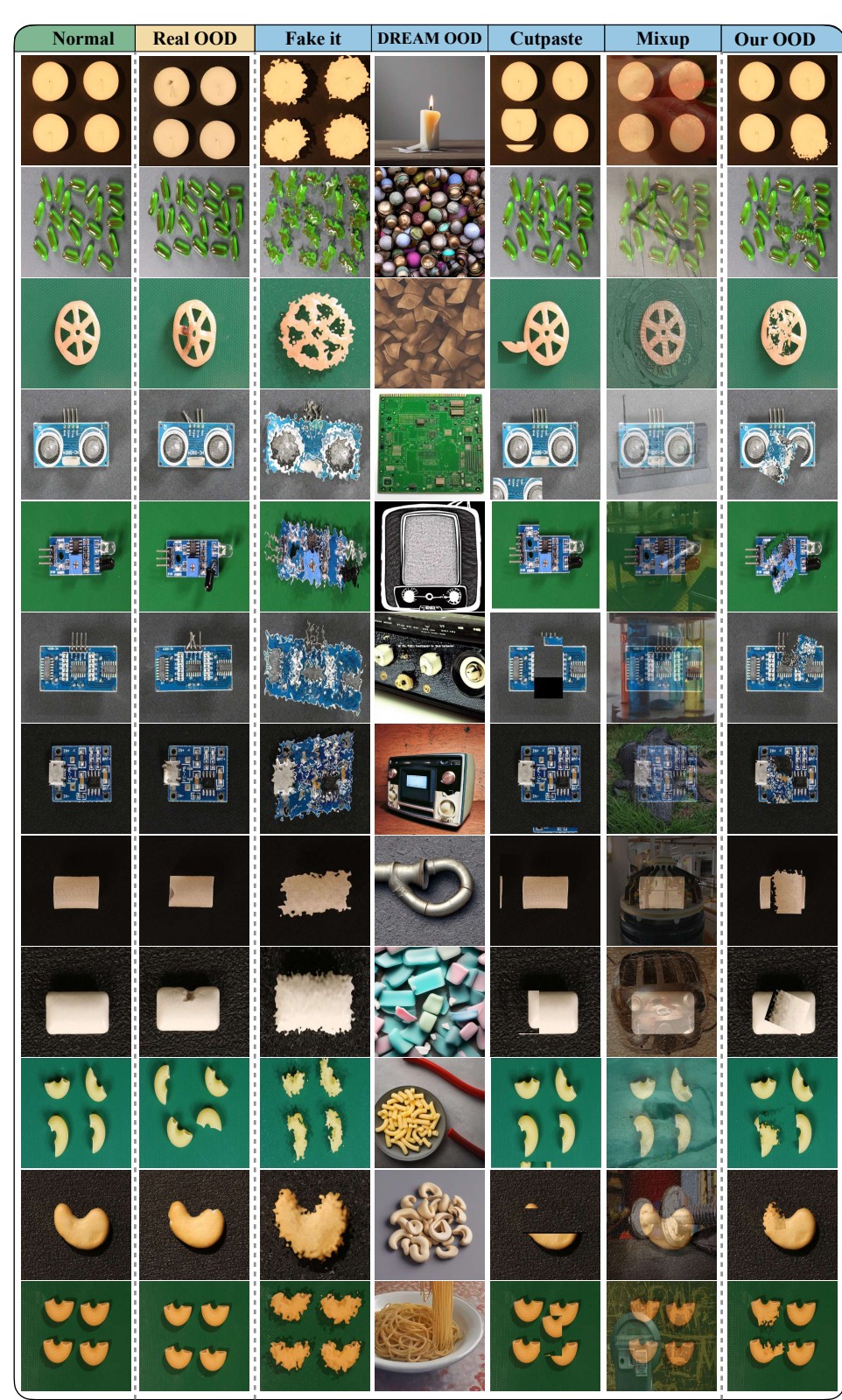

Figure 11: **OOD Generator methods comparison on the VisA dataset.**

