# OpenReview forum: "A Contrastive Teacher-Student Framework for Novelty Detection under Style Shifts"
_ICLR.cc/2025/Conference — Submitted to ICLR 2025_

### Official Review · Reviewer_iu4w · 2024-10-30

**Soundness:** 2
**Presentation:** 2
**Contribution:** 1
**Rating:** 3
**Confidence:** 5

**Summary:**

This paper focuses on the problem of novelty detection under style shifts. And this paper proposes a ND method that crafts an auxiliary OOD set with style features similar to the ID set but with different core features. Then, a task-based knowledge distillation strategy is utilized to distinguish core features from style features. In essence, the performance of the proposed method mainly rely on the quality of the generated data. And this paper only utilizes some commonly-used operations and does not propose any inspired ideas.

**Strengths:**

Novelty detection under style shifts is an important problem. Employing data augmentation is a reasonable solution.

**Weaknesses:**

1. The motivation of this paper is not clear. This paper aims to address novelty detection under style shifts, which involves two shift problems, i.e., covariate shift and semantic shift. However, this paper does not sufficiently analyze why existing methods could not solve these two shifts simultaneously. To the best of my knowledge, there exist some methods that aim to leverage large-scale models, e.g., CLIP, to solve this challenge. The authors should introduce these methods and make an analysis.

2. In Introduction Section, it is better to show a figure to analyze the corresponding problems of existing methods. Besides, I am not clear why the proposed method could solve these two shift problems. The authors should give more interpretations.

3. The proposed method involves data generation and other multiple operations, e.g., contrastive learning. In essence, these operations are commonly used methods and this paper does not propose any inspired ideas, which lacks novelty.

4. In the experiments, the authors should compare more state-of-the-art methods. The testing datasets are somewhat small. The authors should verify their method on more dataset. Besides, the authors should give some feature-level visualization analysis, which is better to understand the proposed method.

**Questions:**

See the weakness.

---

> ### Author Response · Authors · 2024-11-23
>
> Dear Reviewer iu4w,
>
>  Thanks for your constructive and valuable comments on our paper. Below, we provide a detailed response to your questions and comments. If any of our responses fail to address your concerns sufficiently, please inform us, and we will promptly follow up.
>
> >**W1-a:**
>
> As the reviewer noted that the motivation is unclear, we have provided a brief description of the problem setup for the task and clarified our motivation.
>
>
>
> ## **Novelty Detection**
>
> We would like to clarify potential misunderstandings regarding the problem we addressed and the proposed method. To provide clarity, we first review the setup of our problem.
>
> The Novelty Detection (ND) problem involves a scenario where one class is designated as the **in-distribution (ID)** semantic, while all other semantics are considered as out-of-distribution (OOD). The primary goal in novelty detection is to develop a detector model, denoted as **$f$**, to distinguish between ID and OOD concepts.
>
> To formalize this task, a detector model $f$ and a dataset $D$ are considered, where one class, such as $X$, is designated as the ID class, while the remaining classes, $D \setminus X$, constitute the OOD samples. A superior detector is one that more effectively distinguishes ID (i.e., $X$) from OOD (i.e., $D \setminus X$).
>
> The difference between $X$ and $D \setminus X$ is based on a set of features, which we categorize into two types: **core features** and **style feature**. Core features capture the essential semantic differences between ID and OOD, while style features represent non-essential differences that do not generalize well across datasets. In this study, we aim to focus on learning core features and disregard style features.
>
> For example, consider a specific subset of the colored MNIST dataset, $D$, where $X$ consists of green images of the digit 0'' (ID), and $D \setminus X$ consists of red images of the digit 1'' (OOD). The primary core feature here is the distinction between the digits 0'' and 1.'' However, there is also a style feature difference---namely, the color of the digits. A detector could distinguish ID from OOD based on either core features (digit identity) or style features (color).
>
> In this study, our objective is to learn core features, as they are more robust and transferable to new datasets. For instance, consider a new dataset $D'$ during inference, where $X'$ consists of blue images of the digit 0'' (ID), and $D' \setminus X'$ consists of blue images of the digit 1'' (OOD). Here, the style feature (color) remains the same, and a model that relied on style features for distinguishing ID from OOD would perform no better than random guessing. In contrast, a model that learned core features would still perform effectively, as the core feature (digit identity) remains consistent.

---

> > ### Author Response · Authors · 2024-11-23
> >
> > >**W1-b:**
> >
> > ## **Motivation**
> > ### Why Developing a Robust ND Method is Important?
> >
> > Robust ND is critical in various real-world scenarios where environmental variations can lead to style shifts. Below are some examples:
> >
> > ### 1. Autonomous Driving
> > - **Scenario**: An ND model trained on images of roads in one city (e.g., Berlin) might encounter roads in another city (e.g., Los Angeles) with different lighting conditions, weather, or architectural styles.
> > - **Importance**: The system must reliably distinguish unusual objects like pedestrians or animals on the road (novelties) regardless of the style differences, such as sunny versus rainy conditions.
> >
> > ### 2. Medical Imaging
> > - **Scenario**: Medical images such as MRI or CT scans might be captured using different equipment or imaging protocols across hospitals, leading to style shifts in the data.
> > - **Importance**: Detecting anomalies like tumors or lesions should rely on core pathological features rather than stylistic variations introduced by imaging devices or techniques.
> >
> > ### 3. Industrial Quality Control
> > - **Scenario**: Automated inspection systems in factories may analyze products under different lighting conditions or camera settings.
> > - **Importance**: The system must detect defective products or anomalies regardless of changes in visual style caused by environmental or equipment variations.
> >
> > ### 4. Video Surveillance
> > - **Scenario**: Surveillance systems deployed across different locations or times of day may face variations in background, lighting, or weather conditions.
> > - **Importance**: Detecting suspicious activities or objects should remain unaffected by these style shifts, ensuring consistent performance in diverse settings.
> >
> > ### 5. Wildlife Monitoring
> > - **Scenario**: Cameras deployed in different ecosystems or under varying weather conditions may capture images with substantial stylistic differences.
> > - **Importance**: Identifying new species or unusual animal behavior requires robustness to such style shifts.
> >
> > ### 6. Retail and E-commerce
> > - **Scenario**: ND models used to monitor inventory might encounter different lighting, packaging designs, or shelf arrangements across stores.
> > - **Importance**: Detecting misplaced or counterfeit items should not depend on these stylistic changes.
> >
> > ### 7. Satellite and Aerial Imaging
> > - **Scenario**: Satellite images of the same location might appear different due to atmospheric conditions, seasons, or times of day.
> > - **Importance**: Detecting deforestation, urban development, or natural disasters requires focusing on core changes rather than irrelevant stylistic variations.
> >
> > ### 8. Cybersecurity
> > - **Scenario**: Network traffic data might vary in structure due to changes in protocols or encryption methods.
> > - **Importance**: Robust ND is essential to detect novel cyber-attacks while ignoring benign variations in network activity style.

---

> ### Author Response · Authors · 2024-11-23
>
> >**W1:**
>
> >**The motivation of this paper is not clear**
>
> We kindly hope that reviewing the problem setup and the highlighted real-world examples provides clarity on why the robust ND problem serves as a strong motivation for our work.
>
>
> >**This paper aims to address novelty detection under style shifts, which involves two shift problems, i.e., covariate shift and semantic shift. However, this paper does not sufficiently analyze why existing methods could not solve these two shifts simultaneously....**
>
>
> We believe that one of the key limitations of existing ND methods in achieving robustness stems from their reliance on strong inductive biases tailored to specific datasets. SOTA methods for these datasets are often designed with assumptions that align closely with the dataset's unique characteristics. For instance, PatchCore [1], which achieves an impressive 99.6% AUROC on the MVTecAD dataset, relies heavily on patch-based feature extraction. While this approach performs exceptionally well on datasets like MVTecAD, which primarily feature texture-based novelty samples (e.g., a broken screw versus an intact screw), it tends to degrade in performance on datasets emphasizing semantic novelty detection (e.g., distinguishing a dog as a novel concept when a cat is considered inlier).
>
> This limitation is also evident in the standard deviation of performance metrics, where our method consistently demonstrates lower variance compared to existing approaches (Please refer to Table 18.). This suggests that our method is inherently more robust across varying novelty detection datasets.
>
>
>
> >**..To the best of my knowledge, there exist some methods that aim to leverage large-scale models, e.g., CLIP, to solve this challenge. The authors should introduce these methods and make an analysis.**
>
>
>
> There are numerous studies that explore the robustness of CLIP models [2,3,4]; however, we did not find specific works demonstrating the inherent robustness of CLIP for the ND problem.
>
> Although we compared our method to several existing ND approaches, to address the reviewers' concerns, we also compared our method against existing CLIP-based ND methods. The results are summarized in the table below:
>
>
>
> |               | Autonomous Driving | Camelyon17 | Brain Tumor | Chest CT-Scan | W. Blood Cells | Skin Disease | Blind Detection | MVTec AD | VisA |
> |-|-|-|-|-|-|-|-|-|-|
> | CLIP-AD  [5]     | 86.7 / 73.2 | 72.1 / 64.6 | 89.2 / 73.8 | 69.2 / 60.3 | 86.3 / 67.7 | 86.2 / 65.1 |  83.2 / 65.1 |  76.2 / 65.0 | 74.3 / 59.0 |
> | WinCLIP [6]      | 87.2 / 74.6 | 72.9 / 66.7 | 86.6 / 72.8 | 70.2 / 61.7 | 85.7 / 66.8 | 83.3 / 66.8 |  86.9 / 66.1 |  91.8 / 69.1 | 78.1 / 65.8 |
> | AnomalyCLIP  [7]  | 88.0 / 75.2 | 73.4 / 69.3 | 90.3 / 74.8 | 71.8 / 65.9 | 87.4 / 68.0 | 89.7 / 68.5 |  87.9 / 67.1 |  91.5 / 68.5 | 82.1 / 67.1 |
> | OUR           | 92.9 / 84.2 | 75.0 / 72.4 | 98.2 / 79.0 | 72.8 / 71.6 | 88.8 / 72.1 | 90.7 / 70.8 |  96.1 / 73.2 |  94.2 / 87.6 | 89.3 / 82.1 |
>
>
> The results clearly indicate that our method outperforms these competing approaches across all datasets.
>
>
>
> [1] Karsten Roth, Towards Total Recall in Industrial Anomaly Detection
>
> [2] Mitigating Spurious Correlations in Multi-modal Models during Fine-tuning
>
> [3] Fairness and Bias in Multimodal AI: A Survey
>
> [4] MM-SpuBench: Towards Better Understanding of Spurious Biases in Multimodal LLMs
>
> [5] Exposing Outlier Exposure: What Can Be Learned From Few, One, and Zero Outlier Images 2022
>
>
> [6] WinCLIP: Zero-/Few-Shot Anomaly Classification and Segmentation 2023
>
>
> [7] ANOMALYCLIP: OBJECT-AGNOSTIC PROMPT LEARNING FOR ZERO-SHOT ANOMALY DETECTION 2024

---

> ### Author Response · Authors · 2024-11-23
>
> >**W2-a:**
>
> >**In Introduction Section, it is better to show a figure to analyze the corresponding problems of existing methods:**
>
> We kindly request the reviewer to refer to Figure 1, provided immediately after the Introduction, where we analyze the limitations of existing methods in detail.
>
> >**Besides, I am not clear why the proposed method could solve these two shift problems. The authors should give more interpretations.**
>
> Shift Problem 1: Performance drops when test data exhibits style shifts (e.g., environmental variations).
>
> Shift Problem 2: Unwanted correlation between style features and labels due to training data limitations (only ID samples available).
>
> **Understanding the Shift Problems**
>
> Shift Problem 1: Traditional ND methods assume that the training and test data come from the same environment and share similar style features (like lighting, texture, or color). However, in real-world applications, test data often have style variations not present in the training data. For example, images taken under different lighting conditions or with different camera settings. These style shifts can cause existing models to misclassify ID samples as OOD because the models have learned to associate specific style features with the ID class.
>
> Shift Problem 2: Since only ID samples are available during training, models can inadvertently learn to rely on style features present in the ID data as cues for classification. This leads to a spurious correlation where the model associates the presence of certain style features with the ID label. Consequently, if an OOD sample shares these style features, it might be incorrectly classified as ID, and if an ID sample has different style features, it might be misclassified as OOD.
>
> **How Our Proposed Method Addresses These Shift Problems**
>
> 1. Crafting an Auxiliary OOD Set by Distorting Core Features
>
> Identifying Core Features:
>
> * We use feature attribution methods like Grad-CAM to generate saliency maps for ID samples. These maps highlight the core regions of the image that the model considers important for making predictions.
>
> * By applying light augmentations (e.g., slight changes in brightness or contrast) and generating saliency maps for both the original and augmented images, we create a final saliency map that is less sensitive to style features and more focused on core features.
>
> Distorting Core Features:
>
> * We apply hard transformations (e.g., elastic transformations, cutouts) specifically to the core regions identified in the saliency maps.
>
> * This process alters the essential parts of the image that are critical for determining whether it's ID or OOD, effectively creating synthetic OOD samples.
>
> * By doing so, we generate OOD samples that share the same style features as the ID samples but differ in core features.
>
> Why This Helps:
>
> For Shift Problem 1:
>
> By maintaining the style features while altering the core features, we teach the model that style features are not reliable indicators for prediction.
> The model learns to focus on the core features that truly distinguish ID from OOD, making it more robust to style variations in the test data.
>
>
>
> For Shift Problem 2:
>
> * The introduction of OOD samples that share style features with ID samples but differ in core features breaks the spurious correlation between style features and the ID label.
>
> * The model learns that style features alone are insufficient for accurate classification, reducing the reliance on these features.
>
> 2. Applying Style Augmentations to Both ID and Crafted OOD Samples
>
>
> Light Augmentations:
>
> * We apply light augmentations (e.g., color jitter, slight rotations) to both the ID and crafted OOD samples during training.
> These augmentations simulate various style shifts that might occur in real-world scenarios.
> Why This Helps:
>
> For Shift Problem 1:
>
> Exposing the model to a wide range of style variations during training encourages it to become invariant to such shifts.
> The model learns that the label (ID or OOD) remains the same despite changes in style features, reinforcing the focus on core features.
> For Shift Problem 2:
>
> By varying style features while keeping labels consistent, we further weaken any spurious correlations between style features and labels.
> The model is trained to disregard style variations when making predictions.
> 3. Task-Based Knowledge Distillation Framework
> Teacher-Student Model:
>
> We use a pre-trained teacher model (with a trainable binary classification layer) and a student model trained from scratch.
> The teacher is trained to classify the crafted ID and OOD samples, updating only its binary classification layer.
> Novel Loss Function:
>
> We introduce a novel objective function that encourages the student model to align its outputs with the teacher's outputs for ID samples and to diverge for OOD samples.
> This loss function is contrastive, meaning it pulls together representations of similar samples (ID) and pushes apart representations of dissimilar samples (ID vs. OOD).

---

> ### Author Response · Authors · 2024-11-23
>
> >**W2-b:**
>
> Why This Helps:
>
> For Shift Problem 1:
>
> The teacher model, trained on both ID and crafted OOD samples with various style augmentations, learns to focus on core features and ignore style variations.
> The student model, by aligning with the teacher on ID samples regardless of style shifts, also becomes invariant to these shifts.
> For Shift Problem 2:
>
> By forcing divergence on OOD samples that share style features with ID samples, the student model learns that style features are not sufficient for predicting a sample as ID.
>
> The student model is encouraged to rely on core features for making predictions, reducing reliance on spurious correlations.
>
> 4. Causal Viewpoint and Intervention
>
> Breaking Unwanted Correlations:
>
> From a causal perspective, style features act as confounders that can mislead the model.
> By intervening on the core features (altering them to create OOD samples) and varying style features (through augmentations) without changing labels, we weaken the causal link between style features and the label.
> This intervention helps the model focus on the true causal factors (core features) that determine whether a sample is ID or OOD.
>
> Why This Helps:
>
> For Both Shift Problems:
> By disrupting the spurious causal pathways that link style features to the label, the model is less likely to rely on these features.
> The model becomes robust to style shifts and focuses on the features that are truly indicative of the sample's class.
>
> >**W3:**
>
>
>
> We understand the reviewers' concerns regarding the modules in our pipeline, which build upon foundational concepts in the field, such as data-centric approaches and contrastive learning. However, we believe there are several points that highlight the novelty of our study:
>
> * While the mentioned principles are well-established, they remain essential components in Novelty Detection research, as evidenced by recent works such as ReContrast (NeurIPS 2023) and General AD (ECCV 2024).
>
> * Our proposed method makes a distinct contribution by addressing a critical and underexplored challenge in Novelty Detection—achieving robustness under style shifts.
>
> * Our framework employs a causal approach with an effective strategy to craft auxiliary OOD data. Although alternative strategies exist for crafting OOD samples, we differentiate ourselves with a data-efficient approach. Many existing methods rely on large generators (e.g., Stable Diffusion [1,2]) or extensive datasets (e.g., LAION-5B [1,2]); on the other hand, our approach does not require additional datasets. Moreover, our framework is underpinned by a theoretical foundation that validates its design and demonstrates its effectiveness, setting it apart from most existing ND approaches.
>
> * Furthermore, our task-based knowledge distillation strategy goes beyond simply reusing established techniques. By introducing a novel loss function and defining an auxiliary task, the teacher model is first adapted to the ID set and subsequently used to train the student model. This approach ensures alignment on ID samples while encouraging divergence on crafted OOD samples.
>
> * Our approach enhances performance, achieving superior results in both clean and robust evaluations, even on challenging real-world datasets, without relying on metadata or additional datasets.
>
> [1] Du et al, Dream the OOD: Diffusion Models, Neurips 2023
>
> [2] RODEO: Robust Outlier via ICML 2024

---

> ### Author Response · Authors · 2024-11-23
>
> >**W4:**
> We kindly request the reviewer to refer to Table 18, where we have included a comparison with the most recent ND methods. For convenience, we have also included the table here to highlight the superior performance of our method.
>
> >**In the experiments, the authors should compare more state-of-the-art methods.**
>
> |**Method**|**Driving**|**Camelyon**|**Brain**|**Chest**|**Blood**|**Skin**|**Blind**|**MVTec**|**VisA**|**Waterbirds**|**DiagViB-FMNIST**|**Avg↑**|**Clean.Std↓**|
> |-|-|-|-|-|-|-|-|-|-|-|-|-|-|
> |SimpleNet|82.6/63.7|64.7/54.5|89.1/60.2|62.4/50.7|61.4/54.1|82.2/64.7|86.7/58.4|99.6/65.1|96.8/71.0|68.1/59.8|78.8/58.3|79.3/60.0|13.5|
> |DDAD|86.4/65.2|65.3/59.7|90.9/61.4|60.2/45.8|60.9/52.7|84.2/65.1|91.8/57.1|99.8/62.6|98.9/60.4|64.8/58.7|76.5/61.6|80.0/59.1|15.1|
> |EfficientAD|86.1/70.1|68.4/59.6|91.5/65.7|61.9/52.2|63.7/54.3|86.7/63.4|88.6/60.3|99.1/59.7|98.1/57.5|65.7/59.1|78.3/59.4|80.7/60.1|13.8|
> |DiffusionAD|84.8/61.9|67.6/63.4|88.7/63.3|63.0/54.8|60.2/56.1|85.7/64.0|87.3/61.7|99.7/67.1|98.8/63.8|66.8/63.1|75.8/60.7|79.9/61.8|14.0|
> |ReconPatch|83.9/69.3|68.0/56.9|87.6/59.6|62.8/55.1|55.9/53.7|64.0/63.1|89.7/57.6|99.6/60.2|95.4/61.2|65.0/60.5|76.8/59.3|77.2/59.7|14.9|
> |GLASS|85.3/66.7|68.1/57.4|90.4/63.7|63.7/57.7|63.5/54.1|87.2/62.7|90.3/60.7|99.9/65.3|98.8/62.7|68.4/61.7|79.7/63.7|81.4/61.5|13.6|
> |GeneralAD|89.5/73.9|69.1/64.2|91.4/71.0|64.5/62.7|65.7/63.1|89.7/66.4|88.3/57.1|99.2/67.2|95.9/64.9|70.3/65.7|78.3/64.7|82.0/65.5|12.7|
> |GLAD|89.7/70.1|70.5/62.9|90.8/68.4|65.9/61.9|64.9/59.5|90.0/65.7|91.8/58.7|99.3/63.7|99.5/60.4|71.8/63.7|80.9/60.9|83.2/63.3|12.9|
> |**Ours**|92.9/84.2|75.0/72.4|98.2/79.0|72.8/71.6|88.8/72.1|90.7/70.8|96.1/73.2|94.2/87.6|89.3/82.1|76.5/74.0|92.1/78.7|**87.9/76.9**|**8.9**|
>
>
> >**The testing datasets are somewhat small. The authors should verify their method on more dataset.**
>
> Our experiments utilize a diverse set of large-scale datasets, encompassing both real-world and synthetic distribution shifts, as detailed in Appendix J. Specifically, we include datasets from various domains such as autonomous driving (Cityscapes, GTA5), medical imaging (Camelyon17, Brain Tumor, Blindness Detection, Skin Disease, Chest CT-Scan, White Blood Cells), and industrial anomaly detection (MVTecAD, VisA). We are also open to evaluating our method on any dataset the reviewer suggests.
>
>
> >**Besides, the authors should give some feature-level visualization analysis, which is better to understand the proposed method.**
>
>
> As our method operates in the image domain, we have included several images to provide intuition about its functioning. Additionally, to address the reviewer's concerns, we have provided feature space visualizations in Appendix P of the revision. Any additional details or suggestions regarding the visualizations would be greatly appreciated to help us create more informative and insightful plots.

---

> > ### Comment · Reviewer_iu4w · 2024-11-25
> >
> > After reading this paper and the reply, I still consider that this paper does not provide any inspired idea. For novelty detection under styple shifts, it belongs to a new setting that involves two different shift cases, i.e., semantic and covariate shifts. To this end, the authors do not sufficiently analyze the corresponding challenges. Meanwhile, contrastive learning, data generation, and knowledge distillation are three commonly-used methods. We can utilize the combination of the three strategies to process any OOD settings. However, I can not obtain any inspired idea. Which factors lead to OOD problem?  Can we utilize a new different mechanism to address all OOD scenarios?

---

> ### Author Response · Authors · 2024-11-25
>
> First, note that as mentioned in the introduction lines 53 and 54, existing methods mostly require accessing either class labels, or else environment labels (also known as group labels in the spurious correlation mitigation setup) to work. Unfortunately, this information is unavailable in the novelty detection setup, where only the unlabeled samples from the normal class are given.
> To clarify, in our setup, we assume that the features consist of two parts, "core" denoted by $x_c$, and "environment related" denoted as $x_e$. We assume that $x_c$ would only causally affect the label, i.e. normal and anomaly (see fig 2a). Therefore, we tested against whether an ND algorithm could successfully classify normal vs. anomaly when $x_e$, or its distribution, changes at the test time.
> Many existing methods rely on two sets of augmentations ($T^+$ and $T^-$ known as light and hard augmentations) to learn a new representation that is invariant against natural variations *within the normal class*, but be sensitive to the changes that make a normal sample anomalous.
> These methods suffer from the fact that the augmentation is applied to the *entire* feature set, i.e. both $x_e$ and $x_c$. We argue that it is essential to *only* apply the hard augmentation to the core part $x_c$, as mentioned in lines 297 and 298. The reason is that the training data is often associated with a spurious correlation between $x_e$ and $y$, therefore, the learning algorithm may wrongly capture the variations in $x_e$ as signs of anomaly. To break this correlation, we suggested to *only* apply the hard augmentation on $x_c$ to obtain synthetic OODs while keeping $x_e$ unchanged. This helps in breaking the spurious correlation between $x_e$ and $y$, and preventing the algorithm to learn this spurious connection. All this analysis is provided in Sec. 4 Causal Viewpoint.
> Therefore, we believe that making a distinction between $x_c$ and $x_e$ in applying hard augmentations to make synthetic OODs is the major distinction between our method and prior algorithms, and potentially the reason why our algorithm works in our setup. The results also backup this hypothesis: looking at the table 3 in the ablation studies, and comparing setup D, in which "core estimation" is replaced with some random selection of the image patch to be augmented, against setup E, which is our proposed method, we see a big jump in the robust AUROCs.

---

### Official Review · Reviewer_LwHG · 2024-10-30

**Soundness:** 2
**Presentation:** 3
**Contribution:** 3
**Rating:** 6
**Confidence:** 4

**Summary:**

The manuscript proposes a method that addresses the issue of style variations in novelty detection by creating auxiliary OOD sets combined with a task-oriented knowledge distillation strategy. This approach enhances the model's robustness by generating OOD samples through the identification and distortion of core features. However, the method not only requires the use of saliency methods to identify key objects in images but also necessitates the application of hard transformations to regions with high saliency values. This adds significant detection costs and poses challenges for direct application in certain segmentation tasks, which may limit the method's practical usability in real-world scenarios. Additionally, while task-based knowledge distillation strategies have already been applied in some OOD detection tasks, the innovation of the proposed method is somewhat limited. The core idea remains focused on mitigating the impact of different styles on OOD detection. However, in real-world applications, variations in style as well as changes between ID and OOD categories can lead to significant fluctuations in performance. It is crucial for the model to accurately identify OOD categories not only under a single style but also across various style scenarios, such as sunny and rainy conditions. This broader applicability holds substantial research value.

**Strengths:**

The manuscript proposes a novel novelty detection method that combines data augmentation with a knowledge distillation strategy. By integrating a saliency detection task, the method effectively improves detection accuracy and validates its effectiveness across various datasets, particularly in some medical imaging datasets, which adds significant research value. The overall experimental results are comprehensive, and the method description is relatively clear.

**Weaknesses:**

1.	The method relies on saliency detection to identify key objects in the image and applies hard transformations to regions with high saliency values. This may lead to significant computational and time costs. In practical applications, such costs could limit its widespread use, especially in detection and segmentation tasks where multiple OOD key objects are present in the scene, making direct application impractical.
2.	Style Variation in Real-World Scenarios: The core idea of the paper is to mitigate the impact of different styles on OOD detection. However, style variations in real-world scenarios are complex, and changes between ID and OOD can significantly affect model performance. The research value of the model lies in its ability to accurately identify OOD categories across various stylistic contexts, such as sunny and rainy conditions.
3.	Currently, task-based knowledge distillation strategies have been applied in several OOD detection tasks. The method does not clearly demonstrate how it differs from other approaches that utilize knowledge distillation. The manuscript may need to further elaborate on the innovative aspects of this approach.

**Questions:**

1. The manuscript needs to further clarify the detection costs associated with creating auxiliary OOD sets and the limitations it poses for downstream OOD detection and segmentation tasks.
2. The manuscript needs to further articulate the necessity and innovation of the proposed task-oriented knowledge distillation in comparison to other knowledge-based methods for novelty detection.

---

> ### Author Response · Authors · 2024-11-23
>
> Dear Reviewer LwHG,
>
> Thank you for your review and useful comments. Specific comments are answered below:
>
> >**W1&Q1:**
>
>
> We appreciate the reviewer’s observation regarding potential computational and time costs. Below, we address this concern with detailed points supported by empirical data.
>
> 1. Saliency Detection Occurs Only During Training
>
> The saliency detection process is limited to the training phase, where it is used to craft auxiliary OOD samples. This means there is no computational overhead during inference, where real-time performance is critical.
>
> Once the model is trained, saliency detection is no longer needed, ensuring that the inference process remains as efficient as standard methods.
>
>
>
> 2. Efficiency of Grad-CAM
>
> Our method employs Grad-CAM, a lightweight and computationally efficient approach for saliency detection. Compared to other saliency methods, Grad-CAM introduces minimal overhead while generating effective saliency maps.
>
>
>
> 3. One-Time Computation Per Sample
>
> Saliency maps are computed once per training sample and reused throughout the augmentation pipeline. This one-time computation avoids repetitive processing, keeping the overall training cost manageable.
>
>
>
> 4. Empirical Evidence of Training Overhead
>
> Based on time complexity analysis of our method across various datasets in below table. Grad-CAM computation time is negligible and insignificant compared to the training and testing times, making it efficient for real-world applications.
>
> | Dataset              | Grad-CAM (h) | Training (h) | Testing (h) | Shifted Testing (h) |
> |-|-|-|-|-|
> | **Autonomous Driving** | 0.1         | 6.4          | 0.2         | 0.2                 |
> | **Camelyon**          | 1.6         | 279.4        | 0.8         | 0.6                 |
> | **Brain Tumor**       | 0.1         | 1.3          | 0.1         | 0.1                 |
> | **Chest CT-Scan**     | 0.1         | 18.5         | 0.1         | 0.1                 |
> | **White Blood Cells** | 0.1         | 0.2          | 0.1         | 0.1                 |
> | **Skin Disease**      | 0.1         | 8.6          | 0.1         | 0.1                 |
>
>
> The table shows that the additional computational cost during training is insignificant. Considering the performance improvements demonstrated in our work, this overhead is justifiable for practical applications.
>
>
> We believe these clarifications address the reviewer’s concern. By emphasizing that saliency detection is confined to the training phase, leveraging the lightweight Grad-CAM approach, and demonstrating modest computational overhead with empirical data, we establish that the proposed method remains practical and efficient for real-world use cases.

---

> ### Author Response · Authors · 2024-11-23
>
> >**W2:**
>
> We appreciate the referee's feedback regarding the complexity of style variations in real-world scenarios and their impact on OOD detection, a point with which we fully agree. We would like to emphasize that the style shifts introduced during the training phase of our method are intentionally limited and minor. However, our proposed pipeline is evaluated extensively on complex, unseen style shifts during testing, highlighting its effectiveness and robustness.
>
> To address the referee's concerns, we would like to further clarify that the test datasets analyzed in our paper encompass a broad spectrum of both synthetic and natural style variations, as detailed in Section J of the Appendix:
>
> **Synthetic Style Variations**
>
> * In datasets such as DiagViB-MNIST, we introduced artificial changes, including variations in texture, brightness, saturation, and spatial placement. These synthetic alterations are designed to emulate diverse stylistic shifts that can occur in real-world data, providing a controlled environment to rigorously evaluate the model’s robustness.
>
> **Natural Style Variations**
>
> * Autonomous Driving: We utilized large-scale datasets such as Brain Tumor, Chest CT-Scan, Cityscapes, and GTA5, which exhibit significant stylistic differences. For instance, Cityscapes captures German streets, while GTA5 represents synthetic U.S. streets, introducing natural variations in lighting conditions, road markings, and atmospheric effects. These datasets test the model's adaptability to a wide range of environmental and stylistic shifts. Camelyon-17: This dataset presents inter-hospital variations, arising from differences in imaging equipment, protocols, and lighting conditions. These variations realistically simulate the style shifts commonly encountered in medical imaging tasks, challenging the model to generalize features across diverse sources.
> By considering these datasets, we ensure that the model is rigorously tested across a wide spectrum of stylistic variations, ranging from synthetic alterations to realistic environmental and procedural changes. Moreover, our methodological framework is designed to capture core features that are invariant to style shifts, irrespective of their complexity.
>
> While we acknowledge that real-world style variations can extend beyond those examined in our study, the datasets we selected provide a comprehensive and diverse set of challenging conditions to evaluate the robustness of our proposed method.
>
> Finally, we wish to emphasize that our study does not claim to have fully resolved the robust ND problem. Instead, our goal is to highlight the critical importance of this issue in real-world scenarios and propose a theoretically grounded method that achieves superior performance compared to existing approaches, as demonstrated in both clean and robust settings. We view our work as the beginning of a longer research trajectory, not the conclusion of the journey.

---

> ### Author Response · Authors · 2024-11-23
>
> **Q2&W3:**
>
> Several methods based on knowledge distillation have been proposed for novelty detection tasks, and this remains an active area of research, as evidenced by works cited in references [1-7]. While our results on multiple challenging real-world datasets underscore the effectiveness of our approach, we believe our method introduces critical innovations that set it apart from existing knowledge distillation-based approaches:
>
> **Task-Based Knowledge Distillation:**
>
> We propose a novel task-driven knowledge distillation pipeline that classifies samples into two categories—'inliers' and 'outliers'—to learn the inlier distribution while incorporating additional information to selectively update parts of the teacher model’s weights. This approach diverges from conventional methods, which rely solely on the pretrained teacher's weights and are prone to biases originating from the dataset used for initial training. In contrast, our method integrates a task-specific objective, overcoming the limitations of existing approaches whose objective functions merely aim to mimic the teacher's weights without leveraging any auxiliary task.
>
> **Enhanced Loss Function for Robustness:**
>
> We introduce a new loss function designed not only to encourage the student model to mimic inlier features but also to actively diverge from outlier features. This augmentation significantly improves the robustness of the pipeline in handling distribution shifts—an aspect not addressed in prior knowledge distillation-based methods.
>
> State-of-the-Art Robustness:
> Unlike previous works that primarily focus on traditional novelty detection benchmarks, our study highlights how distribution shifts can degrade the performance of existing methods. Our proposed model not only achieves state-of-the-art performance on standard benchmarks but also demonstrates superior robustness to distribution shifts. Please kindly refer to Table 1 for a comparison of average performance under clean and robust setups.
>
> **Theoretical Insights via Causal Perspective:**
>
> Our work also offers theoretical insights by adopting a causal viewpoint to identify and mitigate unwanted correlations between style features and labels—issues that often undermine the effectiveness of novelty detection methods. This theoretical framework supports the design of our task-based knowledge distillation strategy, enabling the model to focus on core features and enhancing robustness against style variations without the need for auxiliary datasets or metadata.
>
> **Extensive Ablation Studies:**
>
> We conducted extensive ablation studies on prior teacher-student methods, analyzing their loss functions (Appendix H) and pipelines (Section 7). The results demonstrate the significant superiority of our method over previous approaches.
>
> **References:**
>
> [1] Bergmann et al. Uninformed Students: Student–Teacher Anomaly Detection with Discriminative Latent Embeddings. CVPR 2020.
>
> [2] Salehi et al. Multiresolution Knowledge Distillation for Anomaly Detection. CVPR 2021.
>
> [3] Deng et al. Anomaly Detection via Reverse Distillation from One-Class Embedding. CVPR 2022.
>
> [4] Cohen et al. Transformaly - Two (Feature Spaces) Are Better Than One. CVPR 2022.
>
> [5] Guo et al. ReContrast: Domain-Specific Anomaly Detection via Contrastive Reconstruction. NeurIPS 2023.
>
> [6] Cao et al. Anomaly Detection under Distribution Shift. ICCV 2023.
>
> [7] Wang et al. Student-Teacher Feature Pyramid Matching for Anomaly Detection. 2022.

---

> ### Author Response · Authors · 2024-11-26
>
> Dear Reviewer LwHG,
>
> thank you again for your review. We wanted to check in to see if there are any further clarifications we can provide. We hope that our updated PDF, including new experiments and explanations, effectively addresses your concerns.
>
> Best,
> the authors

---

> > ### Comment · Reviewer_LwHG · 2024-11-27
> >
> > Regarding W1, the authors have addressed my concerns well through relevant experimental settings and time complexity analysis. Regarding whether style variations would affect the original detector, the authors explain that they introduce artificial changes, including texture variations, brightness, saturation, and spatial positioning, to simulate style shifts. However, it remains unclear whether these variations might further impact the core features. Since the essence of the paper is to generate auxiliary OOD features by perturbing the core features, the authors need to provide further clarification on this aspect. Despite this, the authors have resolved most of my concerns, so I have decided to change my rating.
> > Final Rating: 6: marginally above the acceptance threshold

---

> > > ### Author Response · Authors · 2024-12-01
> > >
> > > Dear Reviewer LwHG,
> > >
> > > Thank you for your invaluable feedback. We are pleased that your concerns have been addressed.
> > >
> > >
> > > Regarding the remaining question, we understand your concern about whether the style variations introduced in our method might inadvertently affect the core features. To address this comprehensively, we would like to provide a detailed clarification on this aspect.
> > >
> > > **1. Purpose of Style Variations:**
> > >
> > > The style variations we introduce—such as color jitter—are designed to simulate realistic environmental/natural changes that might occur in real-world applications, like changes in lighting, camera settings, or minor positional shifts. These variations aim to challenge the detector model to become invariant to style shifts and focus more on the core features that are essential for distinguishing  ID from OOD samples. By exposing the model to such variations during training, we encourage it to learn representations that are robust to style changes, ensuring reliable performance when encountering style shifts in practical scenarios.
> > >
> > > **2. Preservation of Core Features through Light Augmentations:**
> > >
> > > We carefully select light augmentations, denoted as  $\tau^+$, which are commonly used in self-supervised learning literature  and have been shown to preserve the semantic content of images [1,2,3]. Examples include small changes in brightness, contrast, saturation, hue, and slight geometric transformations like minor translations. These augmentations are intentionally mild to ensure that the core features remain unaffected, allowing the model to learn robust representations that are invariant to such style changes.  Through the applyinh of these light augmentations, the integrity of the core features is maintained while variability in style is introduced, aiding in the training of a detector that relies on core features for prediction.
> > >
> > >
> > >
> > >
> > > **3. Saliency Maps and Focus on Core Features:**
> > >
> > > When computing saliency maps using Grad-CAM, we apply these light augmentations to the input images and then take the element-wise product of the saliency maps from both the original and augmented images. This process enhances regions that are consistently important across style variations, effectively isolating the core features while diminishing the influence of style-related features. By focusing on areas that remain salient despite style changes, we ensure that the core features are accurately identified and used in subsequent steps. This allows us to pinpoint the essential regions of the image that contribute to the model's decision-making, ensuring that the style variations do not overshadow the core features.
> > >
> > > **4. Controlled Impact of Style Variations:**
> > >
> > > We acknowledge that any transformation might have some effect on the image's features. However, the parameters of our light augmentations are carefully chosen to minimize any impact on core features inspired by [1,2,3]. For instance, brightness and saturation adjustments are kept within small ranges that do not alter the object identity or essential characteristics needed for correct classification. By controlling the extent of these augmentations, we ensure that the core features remain intact and that the model's ability to recognize and utilize these features is not compromised.
> > >
> > > **5. Distinction Between Light and Hard Transformations:**
> > >
> > > A critical aspect of our method is the distinction between light and hard transformations. Light transformations ( $\tau^+$) are used to simulate style shifts without affecting core features, helping the model learn to be invariant to such changes. In contrast, hard transformations ($\tau^-$) are intentionally designed to disrupt core features when applied to the identified core regions, such as through elastic distortions or severe cropping. These transformations have been extensively investigated in various areas of the literature (e.g., self-supervised learning) and have been shown to be harmful for preserving semantics, often resulting in a significant shift from the original transformation [4-14]. By applying hard transformations only to the core regions and not to the style regions, we generate auxiliary OOD samples that share style characteristics with ID samples but differ in core features. This strategy enables the model to distinguish between changes in style and alterations in core features, enhancing its robustness and detection capabilities.
> > >
> > >  [1] Chen et al.SimCLR 2020
> > >
> > > [2] Grill et al.BYOL
> > >
> > > [3] He et al.MOCO
> > >
> > >
> > > [4] Kalantidis et al., Hard, 2020
> > >
> > > [5] Li Cutpaste, 2021
> > >
> > > [6] Sinha Negative Data, 2021
> > >
> > > [7] Miyai Rethinking Rotation, 2023
> > >
> > > [8] Zhang Improving, 2024
> > >
> > > [9] Chen Novelty, 2021
> > >
> > > [10] DeVries Improved, 2017
> > >
> > > [11] Yun Cutmix, 2019
> > >
> > > [12] Akbiyik Data, 2019
> > >
> > > [13] Ghiasi Copy-Paste 2020
> > >
> > > [14] Tack CSI, 2020

---

### Official Review · Reviewer_pgZK · 2024-11-02

**Soundness:** 2
**Presentation:** 3
**Contribution:** 2
**Rating:** 8
**Confidence:** 3

**Summary:**

The paper introduces a novel approach for novelty detection under style shift by creating manual OOD training samples from content distortion. The method uses a saliency detector to distinguish the content part from the style part in the input image, then applies strong augmentations to distort the content in the original image to generate OOD samples. A knowledge distillation network is utilized, where the teacher network consists of a frozen encoder and a binary classification head, and the student network is a fully trainable model. The student network is trained with a contrastive objective, aiming to bring closer the ID features of the student-teacher networks and to push away the OOD features. Experiments have been conducted to demonstrate the effectiveness of the proposed approach.

**Strengths:**

1. The paper is clearly presented, with a good logical flow and well-designed figures.
2. The experiments show a visible improvement in novelty detection under style shift.
3. The paper provides theoretical analysis from a causal perspective.

**Weaknesses:**

1. The generated OOD samples might still contain content (core feature) information about ID samples. From the visualized illustrations, it appears that most core features are preserved even after distortion. I am concerned that treating them as negative samples in a contrastive objective might impact the performance of the ND model.
2. In the student-teacher framework, features from different layers within the pre-trained ResNet networks are extracted for contrastive learning. It has been shown that different layers are associated with processing different levels of features; for example, early layers focus on textures and edges, middle layers focus on local patterns, and deep layers capture high-level semantic features. Within this framework, the ideal generated OOD samples should only differ in semantic features while retaining the same local patterns and other style-associated features. Therefore, I believe the framework’s design is not sufficiently justified from a conceptual perspective.

**Questions:**

See Weaknesses Above.

---

> ### Author Response · Authors · 2024-11-23
>
> Dear Reviewer pgZK,
>
> Thank you very much for your feedback on our paper. Please find our responses below:
>
>  >**W1:**
>
>
> We acknowledge the concern regarding the extent of differences between crafted OOD samples and ID data. However, it is important to note that OOD samples can generally be categorized into two groups: (1) texture-level OOD samples, and (2) semantic-based OOD samples. In the first category, OOD samples exhibit significant differences from ID samples at the texture level while retaining some semantic similarity. In the second category, OOD samples are entirely different from ID samples in terms of semantics.
>
> Notably, many challenging novelty detection benchmarks focus on detecting texture-level OOD samples, as they are more representative of real-world tasks. For example, in industrial production lines, detecting broken devices requires identifying subtle texture changes (e.g., the MVTeCad and Visa datasets). Similarly, in medical imaging tasks, ID and OOD differences often arise due to tumors or regional distortions (e.g., Brain Tumor or Chest CT-Scan datasets). For instance, while a brain image with a tumor may share similar global features with a healthy brain image, it is considered OOD due to its specific regional abnormalities.
>
> In our study, we specifically aimed to craft texture-level OOD samples instead of semantic-based OOD samples for the following reasons:
>
> **Data Efficiency:** Crafting texture-level OOD samples is more data-efficient. Generating new semantic-based OOD samples often requires very large generative models or extensive datasets, as demonstrated by previous methods such as Dream-OOD (Du et al., NeurIPS 2023).
>
> **Practical Usefulness of Near OOD:** Studies have shown that near-OOD samples are more useful than far-OOD samples for many applications (ATOM, Chen et al., 2021; POEM, Ming et al., ICML 2022; VOS, Du et al., ICLR 2022). As discussed in our manuscript, near-OOD samples act as placeholders for inlier boundaries provide effective information for learning robust decision boundary . Although the definition of "near OOD" is still evolving, it is generally agreed that these samples share visual appearance similarities with ID samples but do not belong to the ID class.
>
> As a result, we focused on crafting texture-level OOD samples. By distorting a significant portion of an image (e.g., 25% of an important region), we can shift it from ID to OOD while maintaining some similarity to ID samples. This makes them near-OOD samples, which, according to the above definition, are more useful for practical applications.
>
>
> Finally, to fully address the reviewer concern, we conducted an additional experiment across multiple datasets. In this experiment, we focused on distorting all core regions of the images by identifying contours using Grad-CAM maps and applying distortions to all regions within these contours. The results from this experiment will be included in the revised manuscript, demonstrating how varying distortion levels impact the model's performance and its ability to distinguish between ID and OOD samples. This decrease in performance might be attributed to crafting OOD samples that are too far removed from the ID data. For instance, instead of distorting specific brain regions, we applied distortions to the entire brain region to generate OOD samples.
>
> By replacing our default generation strategy with this approach while keeping all other components fixed, we obtained the following results:
>
>
> |               |MVTec AD| Visa | Brain Tumor | Chest CT-Scan | W. Blood Cells | Skin Disease | Waterbirds |
> |-|-|-|-|-|-|-|-|
> | OUR                     | **94.2 / 87.6** | **89.3 / 82.1** | **98.2 / 79.0** | **72.8 / 71.6**  | **88.8 / 72.1** | **90.7 / 70.8**  | **76.5 / 74.0**  |
> | Full distortion of core | 83.4 / 74.3 | 76.7 / 70.1 | 86.2 / 75.8 | 65.7 / 62.3  | 81.6 / 69.8 | 81.5 / 67.5  | 68.9 / 64.8 |

---

> ### Author Response · Authors · 2024-11-23
>
> >**W2:**
>
>
> As is common in the general knowledge distillation literature, where information from different layers is utilized, we were inspired by this approach and aimed to adopt a similar strategy. However, we understand the reviewer’s concerns and provide the following responses:
>
> Early layers in neural networks typically capture low-level features (e.g., edges, shapes) , while deeper layers focus on high-level semantic information. Our framework, however, is based on style features and core features, and while the intuition regarding layer functionality makes sense, it is important to note that low-level features are not equivalent to style features, nor are high-level features directly equivalent to core features.
>
> For instance, in medical images, ID and OOD differences often revolve around tumor regions, which we consider core features in our study. These features are shape-based and may sometimes appear in shallow layers. Therefore, ignoring shallow layers and relying solely on deeper layers does not seem entirely acceptable. That said, we acknowledge that deeper layers provide meaningful representations and could serve as a good approximation of all layers in certain cases.
>
> To further address the reviewer’s concerns, we conducted additional experiments where we dropped some shallow layers and compared the results with the default settings reported in the manuscript. For these experiments, we now consider four layers and evaluate two configurations:
>
> Config 1: Using only the last two layers.
>
> Config 2: Using the last three layers.
>
>
> |               | Autonomous Driving | Camelyon17 | Brain Tumor | Chest CT-Scan | W. Blood Cells | Skin Disease | Blind Detection | MVTec AD | VisA |
> |-|-|-|-|-|-|-|-|-|-|
> | Config 1       | 89.3 / 81.4 | 73.8 / 69.1 | 96.4 / 75.9 | 71.6 / 69.7 | 85.9 / 70.3 | 87.6 / 65.6 |  94.3 / 70.9 |  90.6 / 84.8 | 86.3 / 80.0 |
> | Config 2       | 90.6 / 83.2 | 74.5 / 70.9 | 97.6 / 77.8 | 72.3 / 70.4 | 87.1 / 71.2 | 88.9 / 68.8 |  95.3 / 72.7 |  92.9 / 86.5 | 87.6 / 81.3 |
> | *Default*       | 92.9 / 84.2 | 75.0 / 72.4 | 98.2 / 79.0 | 72.8 / 71.6 | 88.8 / 72.1 | 90.7 / 70.8 |  96.1 / 73.2 |  94.2 / 87.6 | 89.3 / 82.1 |
>
> These results demonstrate that incorporating shallow layers, even minimally, contributes to performance improvements. However, they also indicate that their role is not pivotal.

---

> ### Comment · Reviewer_pgZK · 2024-11-25
>
> Fair enough. No futher questions on my side. I personally find this paper interesting so I raised my score.

---

> ### Author Response · Authors · 2024-11-25
>
> Dear Reviewer pgZK,
>
>
> Thank you for your valuable review and positive feedback! We are delighted that you found our work interesting.
>
> Sincerely, The Authors.

---

### Official Review · Reviewer_2JvA · 2024-11-03

**Soundness:** 3
**Presentation:** 3
**Contribution:** 2
**Rating:** 5
**Confidence:** 5

**Summary:**

Using knowledge distillation models for anomaly detection, the model tends to learn features related to style, leading to poor performance in cases of style transfer. Therefore, by applying style variations to obtain augmented samples while retaining labels, a more robust representation can be achieved. At the same time, by identifying and distorting the core areas of ID samples through a feature attribution method for data augmentation, unnecessary correlations between style features and labels can be weakened. Minor improvements have been made to the knowledge distillation model part, updating the weights of the binary layer to enhance the teacher's knowledge.

**Strengths:**

1.The ablation study is very comprehensive, and the experimental results and datasets are also abundant.
2.The paper is well-written and organized, with an easy-to-understand approach and a clear presentation of the method.There are various experimental details and pseudocode implementations.

**Weaknesses:**

1.There are no more intuitive experiments to demonstrate that the unwanted correlation between style features and labels has been weakened.
2.The OOD generation strategy heavily borrows from methods in self-supervised learning that preserve and do not preserve semantics, and moreover, there are not many improvements to the knowledge distillation model, which makes the innovation somewhat lacking.Improve the knowledge transfer mechanism in the distillation process, or develop new technologies to more effectively extract knowledge from the teacher model.

**Questions:**

1.If the hyperparameters the parameter alpha have little impact on the experiment, why are they still retained?
2.The proof of the theorem 1 does not seem to effectively demonstrate the effectiveness of the improvement through generating ODD (Out-of-Distribution) samples.

---

> ### Author Response · Authors · 2024-11-23
>
> Dear Reviewer 2JvA,
> Thank you for taking the time to review our paper and for the insightful comments. Here is our detailed response:
>
> >**W1:**
>
> >**There are no more intuitive experiments to demonstrate that the unwanted correlation between style features and labels has been weakened.**
>
>
> we evaluated our method on both synthetic and real-world datasets to rigorously test its ability to focus on core features while being agnostic to style features. For instance, in DiagViB-MNIST, style variations such as brightness, texture, and spatial placement were introduced, and the performance improvements shown in Table 1 demonstrate the model’s robustness in isolating core features. Similarly, the significant performance gains on real-world datasets like Cityscapes and GTA5, which feature large-scale environmental and stylistic shifts, further support our claim. These datasets inherently challenge the model to generalize under varying style conditions, and our method consistently outperforms existing approaches in robust performance metrics (see Tables 1 and 18).
>
>
>
> An additional intuitive experiment showcasing our method's contribution to weakening the correlation between style features and labels is the ablation study (Table 3). Here, we extend the experimental setups to analyze the impact of including or excluding crafted OOD data during training. The results, summarized below, underscore the importance of our approach across diverse dataset, demonstrating notable improvements when OOD data is incorporated.
>
> |               | Autonomous Driving| Camelyon17 | Brain Tumor | Chest CT-Scan | W. Blood Cells | Skin Disease | Blind Detection |
> |-|-|-|-|-|-|-|-|
> | **Default** ( with OOD)    | 92.9 / 84.2  | 75.0 / 72.4  | 98.2 / 79.0  | 72.8 / 71.6  | 88.8 / 72.1  | 90.7 / 70.8  | 96.1 / 73.2  |
> |   without OOD | 81.2 / 65.4 | 66.4 / 57.9  | 91.6 / 54.2  | 64.1 / 58.2  | 80.6 / 63.3  | 73.5 /  61.8 | 84.0 / 57.4|
>
>
> These results showcases the effectiveness of our method in reducing reliance on style features.
>
>
> >**W2:**
>
>
> >**The OOD generation strategy heavily borrows from methods in self-supervised learning that preserve and do not preserve semantics**
>
> Using transformations such  positive augmentation for style shift is a minor aspect of our method, intentionally employed because such augmentations are well-known to preserve semantics.

---

> ### Author Response · Authors · 2024-11-23
>
> **W2:**
>
> >**there are not many improvements to the knowledge distillation model, which makes the innovation somewhat lacking.Improve the knowledge transfer mechanism in the distillation process, or develop new technologies to more effectively extract knowledge from the teacher model.**
>
>
>
> Several knowledge distillation-based methods have been proposed for novelty detection tasks, highlighting this as an active area of research, as demonstrated by references [1-7]. While our results on multiple challenging real-world datasets emphasize the effectiveness of our approach, particularly when compared to knowledge distillation methods such as ReContrast, we believe our method introduces critical innovations that distinguish it from existing knowledge distillation-based techniques.
> **Task-Based Knowledge Distillation:**
>
> We propose a novel task-driven knowledge distillation pipeline that classifies samples into two categories—'inliers' and 'outliers'—to learn the inlier distribution while incorporating additional information to selectively update parts of the teacher model’s weights. This approach diverges from conventional methods, which rely solely on the pretrained teacher's weights and are prone to biases originating from the dataset used for initial training. In contrast, our method integrates a task-specific objective, overcoming the limitations of existing approaches whose objective functions merely aim to mimic the teacher's weights without leveraging any auxiliary task.
>
> **Enhanced Loss Function for Robustness:**
>
> We introduce a new loss function designed not only to encourage the student model to mimic inlier features but also to actively diverge from outlier features. This augmentation significantly improves the robustness of the pipeline in handling distribution shifts—an aspect not addressed in prior knowledge distillation-based methods.
>
> State-of-the-Art Robustness:
> Unlike previous works that primarily focus on traditional novelty detection benchmarks, our study highlights how distribution shifts can degrade the performance of existing methods. Our proposed model not only achieves state-of-the-art performance on standard benchmarks but also demonstrates superior robustness to distribution shifts. Please kindly refer to Table 1 for a comparison of average performance under clean and robust setups.
>
> **Theoretical Insights via Causal Perspective:**
>
> Our work also offers theoretical insights by adopting a causal viewpoint to identify and mitigate unwanted correlations between style features and labels—issues that often undermine the effectiveness of novelty detection methods. This theoretical framework supports the design of our task-based knowledge distillation strategy, enabling the model to focus on core features and enhancing robustness against style variations without the need for auxiliary datasets or metadata.
>
> **Extensive Ablation Studies:**
>
> We conducted extensive ablation studies on prior teacher-student methods, analyzing their loss functions (Appendix H) and pipelines (Section 7). The results demonstrate the significant superiority of our method over previous approaches.
>
> **References:**
>
> [1] Bergmann et al. Uninformed Students: Student–Teacher Anomaly Detection with Discriminative Latent Embeddings. CVPR 2020.
>
> [2] Salehi et al. Multiresolution Knowledge Distillation for Anomaly Detection. CVPR 2021.
>
> [3] Deng et al. Anomaly Detection via Reverse Distillation from One-Class Embedding. CVPR 2022.
>
> [4] Cohen et al. Transformaly - Two (Feature Spaces) Are Better Than One. CVPR 2022.
>
> [5] Guo et al. ReContrast: Domain-Specific Anomaly Detection via Contrastive Reconstruction. NeurIPS 2023.
>
> [6] Cao et al. Anomaly Detection under Distribution Shift. ICCV 2023.
>
> [7] Wang et al. Student-Teacher Feature Pyramid Matching for Anomaly Detection. 2022.

---

> ### Author Response · Authors · 2024-11-23
>
> >**Q1:**
>
> Near-OOD samples, which share similar appearances with ID samples but differ in their core features, are more useful and informative compared to far-OOD samples. These samples provide challenging examples that force the model better delineate the decision boundary between ID and OOD. The mask size parameter $\alpha$ controls the proportion of the image to be distorted, directly impacting the crafting of OOD samples.
>
> As discussed in the manuscript and supported by prior work, selecting an appropriate range for $\alpha$ ensures that the distorted core regions shift the sample away from the ID class while leaving some regions intact to resemble ID samples. This balance is essential to maintain the near-OOD property, allowing the model to focus on meaningful differences while learning robust boundaries. Furthermore, randomizing $\alpha$ increases the diversity of the crafted OOD samples, which enhances robustness by exposing the model to a broader range of scenarios.
>
>  The results in Table 8 (which also provided below) demonstrate the impact of different mask sizes on performance across various datasets.
>
>
> | Mask Size (% of image) | Brain Tumor       | Autonomous Driving | DiagViB-MNIST    | WaterBirds       | MVTec AD         | VISA            |
> |-|-|-|-|-|-|-|
> | 5% to 20%              | 96.2 / 76.1      | 90.0 / 82.1       | 93.0 / 74.2     | 77.0 / 72.3      | 92.1 / 86.7      | 87.8 / 83.0     |
> | 10% to 30%             | 97.1 / 78.9      | 93.0 / 84.3       | 92.5 / 73.2     | 75.0 / 73.9      | 95.1 / 86.4      | 90.1 / 81.5     |
> | 20% to 40%             | 98.3 / 79.4      | 91.3 / 83.8       | 93.4 / 72.1     | 75.4 / 73.1      | 94.3 / 85.1      | 89.7 / 81.2     |
> | 20% to 50% (Default)      | 98.2 / 79.0      | 92.9 / 84.2       | 93.1 / 73.8     | 76.5 / 74.0      | 94.2 / 87.6      | 89.3 / 82.1     |
> | 30% to 50%             | 96.9 / 77.6      | 91.7 / 83.1       | 91.3 / 73.0     | 76.1 / 73.6      | 92.8 / 86.6      | 87.1 / 81.5     |
> | 40% to 70%             | 90.4 / 71.3      | 84.5 / 77.0       | 85.7 / 64.9     | 69.9 / 65.8      | 86.3 / 78.7      | 81.2 / 74.7     |
> | 80% to 100%            | 82.3 / 64.1    | 77.8/ 70.9    | 78.4 / 59.5   | 61.7 / 58.1   | 79.2 /  71.4  | 74.6 / 67.8 |
> | 0% to 100%             |   88.6 / 70.1 | 82.1 / 75.0 | 84.7 / 63.3 | 68.3 / 65.1 | 84.7 / 76.7 | 79.8 / 73.1|
>
> From Table  it is evident that $\alpha$ influences performance. For instance, using a very high mask ratio (e.g., 80\%-100\%) results in far-OOD samples that differ greatly from ID, leading to reduced performance.
>
> Finally, the method demonstrates robustness to minor variations in $\alpha$, as shown by the relatively stable performance across different settings. This indicates the reliability and flexibility of our approach in crafting effective OOD samples.
>
>
> >**Q2:**
>
>
> In Theorem 1, we demonstrated that improving detection performance between ID and real-OOD requires the distribution of synthetic OOD samples to closely resemble that of real-OOD samples. Notably, in many real-world scenarios, ID data is naturally similar to OOD. Thus, achieving this proximity between synthetic and real anomalies can be effectively accomplished through minor distortions of ID data, which is precisely the approach we adopted in our method.

---

> ### Author Response · Authors · 2024-11-26
>
> Dear Reviewer 2JvA,
>
> we have aimed to thoroughly address your comments. If you have any further concerns, we would be most grateful if you could bring them to our attention, and we would be pleased to discuss them.
>
> Sincerely, The Authors

---

> > ### Author Response · Authors · 2024-12-01
> >
> > Dear Reviewer 2JvA,
> >
> > We would like to express our sincere gratitude for your thoughtful review and the valuable insights you've provided on our manuscript. We have carefully considered your feedback and have submitted a revised version of the paper. Your expertise has been instrumental in guiding our revisions, and we are eager to hear your thoughts on the changes we've implemented.
> >
> > We understand that reviewing requires significant effort, and we truly appreciate the time you dedicate to this process. We hope you will be able to review our rebuttal and share any additional comments, which will be essential in further enhancing the quality and clarity of our manuscript. If our clarifications and improvements address your concerns, we would greatly appreciate it if you could reconsider your evaluation.
> >
> > Yours sincerely,
> >
> > The Authors

---

> ### Author Response · Authors · 2024-12-02
>
> Dear 2JvA,
>
> Apologies for reaching out again, but we are keen to hear your feedback before the rebuttal period concludes. Your insights have been incredibly valuable, and we hope the new experiments and revisions to the manuscript allow you to reevaluate your score.
>
> Thank you!

---

> ### Author Response · Authors · 2024-12-03
>
> Dear **Reviewer 2JvA**,
>
> This message serves as a friendly reminder that the discussion period is nearing its conclusion. We have submitted our rebuttal addressing your comments and are hopeful for your feedback. Please let us know if we have successfully addressed your concerns. Should there be any outstanding issues, we are more than willing to continue the dialogue.
>
> Sincerely, The Authors

---

### Official Review · Reviewer_nLRW · 2024-11-03

**Soundness:** 3
**Presentation:** 3
**Contribution:** 2
**Rating:** 6
**Confidence:** 4

**Summary:**

This paper introduces a contrastive teacher-student framework for novelty detection (ND) that improves robustness under style shifts. By generating OOD samples through core feature interventions while preserving style, the method aims to reduce spurious correlations and enhance detection accuracy. Experimental results show significant AUROC improvements, though the focus on style shifts may limit generalizability to other types of distribution changes.

**Strengths:**

+  Innovative Approach to OOD Generation: The paper proposes a novel method for generating out-of-distribution (OOD) samples by intervening in core features of in-distribution (ID) samples. This method leverages saliency maps to identify core regions, making it uniquely capable of creating OOD samples that retain style features but diverge in core attributes.
+ Comprehensive Experimental Results: The paper demonstrates the proposed method's effectiveness across multiple datasets and settings, including autonomous driving and medical imaging, which reinforces the practical applicability of the method in real-world scenarios.

**Weaknesses:**

+ Potential Bias in Robustness Evaluation: The experimental setup for robustness focuses heavily on distribution shifts primarily related to style changes. Since the OOD generation method itself targets this specific type of shift, the evaluation may unfairly favor the proposed method over other ND approaches, which were not designed with style changes as the main concern. This narrow focus on style-based OOD shifts limits the generalizability of the results and may not reflect the performance of the proposed method in more varied OOD scenarios (e.g., class or content-based distribution shifts).

**Questions:**

+ How might the proposed approach perform in scenarios where OOD samples differ from ID samples based on content or class shifts rather than style changes?
+ Have the authors considered conducting additional evaluations using datasets with OOD shifts that are unrelated to style, to better understand the generalizability of the proposed method ？

---

> ### Author Response · Authors · 2024-11-23
>
> Dear Reviewer nLRW,
>
> Thanks for your valuable comments on our paper. Below, we provide a detailed response to your questions and comments:
>
>
> ## **Novelty Detection**
>
> We would like to clarify potential misunderstandings regarding the problem we addressed and the proposed method. To provide clarity, we first review the setup of our problem.
>
> The Novelty Detection (ND) problem involves a scenario where one class is designated as the **in-distribution (ID)** semantic, while all other semantics are considered as out-of-distribution (OOD). The primary goal in novelty detection is to develop a detector model, denoted as **$f$**, to distinguish between ID and OOD concepts.
>
> To formalize this task, a detector model $f$ and a dataset $D$ are considered, where one class, such as $X$, is designated as the ID class, while the remaining classes, $D \setminus X$, constitute the OOD samples. A superior detector is one that more effectively distinguishes ID (i.e., $X$) from OOD (i.e., $D \setminus X$).
>
> The difference between $X$ and $D \setminus X$ is based on a set of features, which we categorize into two types: **core features** and **style feature**. Core features capture the essential semantic differences between ID and OOD, while style features represent non-essential differences that do not generalize well across datasets. In this study, we aim to focus on learning core features and disregard style features.
>
> For example, consider a specific subset of the colored MNIST dataset, $D$, where $X$ consists of green images of the digit 0'' (ID), and $D \setminus X$ consists of red images of the digit 1'' (OOD). The primary core feature here is the distinction between the digits 0'' and 1.'' However, there is also a style feature difference---namely, the color of the digits. A detector could distinguish ID from OOD based on either core features (digit identity) or style features (color).
>
> In this study, our objective is to learn core features, as they are more robust and transferable to new datasets. For instance, consider a new dataset $D'$ during inference, where $X'$ consists of blue images of the digit 0'' (ID), and $D' \setminus X'$ consists of blue images of the digit 1'' (OOD). Here, the style feature (color) remains the same, and a model that relied on style features for distinguishing ID from OOD would perform no better than random guessing. In contrast, a model that learned core features would still perform effectively, as the core feature (digit identity) remains consistent.

---

> ### Author Response · Authors · 2024-11-23
>
> ## **Motivation**
> ### Why Developing a Robust ND Method is Important?
>
> Robust ND is critical in various real-world scenarios where environmental variations can lead to style shifts. Below are some examples:
>
> ### 1. Autonomous Driving
> - **Scenario**: An ND model trained on images of roads in one city (e.g., Berlin) might encounter roads in another city (e.g., Los Angeles) with different lighting conditions, weather, or architectural styles.
> - **Importance**: The system must reliably distinguish unusual objects like pedestrians or animals on the road (novelties) regardless of the style differences, such as sunny versus rainy conditions.
>
> ### 2. Medical Imaging
> - **Scenario**: Medical images such as MRI or CT scans might be captured using different equipment or imaging protocols across hospitals, leading to style shifts in the data.
> - **Importance**: Detecting anomalies like tumors or lesions should rely on core pathological features rather than stylistic variations introduced by imaging devices or techniques.
>
> ### 3. Industrial Quality Control
> - **Scenario**: Automated inspection systems in factories may analyze products under different lighting conditions or camera settings.
> - **Importance**: The system must detect defective products or anomalies regardless of changes in visual style caused by environmental or equipment variations.
>
> ### 4. Video Surveillance
> - **Scenario**: Surveillance systems deployed across different locations or times of day may face variations in background, lighting, or weather conditions.
> - **Importance**: Detecting suspicious activities or objects should remain unaffected by these style shifts, ensuring consistent performance in diverse settings.
>
> ### 5. Wildlife Monitoring
> - **Scenario**: Cameras deployed in different ecosystems or under varying weather conditions may capture images with substantial stylistic differences.
> - **Importance**: Identifying new species or unusual animal behavior requires robustness to such style shifts.
>
> ### 6. Retail and E-commerce
> - **Scenario**: ND models used to monitor inventory might encounter different lighting, packaging designs, or shelf arrangements across stores.
> - **Importance**: Detecting misplaced or counterfeit items should not depend on these stylistic changes.
>
> ### 7. Satellite and Aerial Imaging
> - **Scenario**: Satellite images of the same location might appear different due to atmospheric conditions, seasons, or times of day.
> - **Importance**: Detecting deforestation, urban development, or natural disasters requires focusing on core changes rather than irrelevant stylistic variations.
>
> ### 8. Cybersecurity
> - **Scenario**: Network traffic data might vary in structure due to changes in protocols or encryption methods.
> - **Importance**: Robust ND is essential to detect novel cyber-attacks while ignoring benign variations in network activity style.

---

> ### Author Response · Authors · 2024-11-23
>
> >**Q1:**
>
> As we mentioned in the manuscript, ID and OOD samples differ in core features across **all** considered datasets, and our study aims to develop a robust ND method that learns core features while being agnostic to style features. As noted by the reviewer, core features represent content and semantics. We kindly ask the reviewer’s attention to Figure 4, which includes visualizations illustrating these differences.
>
> For example:
>
> * Brain Tumor Dataset: ID and OOD samples differ based on the presence of a tumor in the brain region.
>
> * Waterbirds Dataset: The distinction between ID and OOD samples lies in the semantics; land birds and water birds are categorized based on their species and context.
>
> * MVTecAD Dataset: ID and OOD samples are differentiated by their condition, where ID samples are intact instances, and OOD samples represent broken devices.
>
> * Colored MNIST Dataset: ID and OOD samples are differentiated similarly based on content and semantics.
>
> These examples, along with the visualizations in Figure 4, demonstrate the relevance of core features and their critical role in robust ND methods.
>
>
> >**Q2:**
>
> We believe that many of the datasets we considered exhibit the mentioned attribute. For instance, in the MVTecAD and VisA datasets, all style features are identical between the ID and OOD samples, while the core features differ and is unrelated to the style. In the mentioned dataset, the core feature is distortion: images without distortion are categorized as ID, whereas those with distortion are categorized as OOD.
>
>
> >**W1:**
>
>
> We understand the reviewers' concern that previous ND methods often aim to improve performance on specific datasets without considering robustness against shifted tasks, leading to unfair comparisons. However, we have corresponding responses to address this concern:
>
> #### Comparison with Existing Methods
>
> Several methods, such as GNL and RedPanda, have aimed to develop robust ND methods targeting problems similar to ours. However, these methods rely on additional supervision to achieve robustness. In contrast, as shown in the results (please refer to Table 1), our method outperforms these approaches.
>
> #### General Robustness and Performance
>
> Our primary focus is on developing a robust ND method rather than merely improving performance on specific datasets. Nonetheless, our method achieves:
>
> - **Higher average performance** across multiple datasets compared to existing ND methods.
> - **Low standard deviation (STD)** across various datasets, demonstrating reliability when applied to new datasets.
> - **Significantly higher robust performance** with a notable margin.
>
> Please refer to **Table 1** and **Table 18** for detailed results.
>
> #### Strengths and Novelty of Our Approach
>
> We acknowledge that part of our method’s robustness stems from incorporating style shifts during training to mitigate the impact of nuisance features during testing. However, our method also introduces several **novel and effective components**, including:
>
> 1. **A novel OOD crafting strategy**
> 2. **A task-based teacher-student pipeline**
>
> These modules collectively enable our model to be robust against a wide variety of natural and synthetic unseen style shifts. It is important to emphasize that the style shifts introduced during training are limited and general (e.g., color jitter). This highlights that our method is not tailored to specific observed style shifts but achieves robustness more broadly.

---

> ### Author Response · Authors · 2024-11-27
>
> Dear Reviewer nLRW,
>
> We sincerely appreciate the thoughtful feedback you’ve offered on our manuscript. We have carefully reviewed each of your comments and have made efforts to address them thoroughly. We kindly ask you to review our responses and share any additional thoughts you may have on the paper or our rebuttal. We would be more than happy to accept all your criticisms and incorporate them into the paper.
>
> Sincerely, The Authors

---

### Meta-Review · Area_Chair_o9tB · 2024-12-13

**Metareview:**

This paper presents a work for novelty detection under style shifts. The core idea is to create OOD samples by distorting content and train a student network via contrastive learning to align ID features with a frozen teacher network while separating OOD features. Experiments demonstrate the superiority of the proposed method in a series of cases. The strengths of this paper include its good organization and presentations. Besides, reported experimental results are promising. However, the technical contributions of this paper are weak and should be improved. The theoretical analysis also should be enhanced to better correspond to the claims. These weaknesses put the work below the acceptance line. The authors can adopt useful comments from reviewers and further polish this work to make a stronger submission.

**Additional Comments On Reviewer Discussion:**

This submission received the comments from five reviewers. Their recommendations are mixed (3 positive and 2 negative). The discussions and changes during the rebuttal period are summarized below.

- Reviewer pgZK provided several questions about OOD sample quality and framework design limitations. The rebuttal convinced the reviewer.
- Reviewer LwHG asked questions about high computational costs, complex real-world style variations, and lack of innovation clarity. The rebuttal handled most of the mentioned concerns. The style variations should be analyzed further in-depth. More explanations and evidence should be provided.
- Reviewer nLRW raised concerns about potential bias in evaluations, limited generalizability, and unaddressed evaluation scenarios. After checking the responses of the authors, the issue of potential bias in evaluations actually should be highlighted and needs more solid explanations (or more clear evidence).
- Reviewer 2JvA was mainly worried about the lack of intuitive validation and limited innovation, and did not reply to the authors. AC checked the questions and rebuttal. The validation is addressed well. However, the technical innovation of this paper is still unclear and should be enhanced by providing more solid evidence.
- Reviewer iu4w initially provided questions about experiments and technical contributions. After rebuttal, the concerns about experiments are addressed mostly. However, the concerns about technical contributions remain, which are also acknowledged in the discussion.

AC appreciates the insights provided by this work. Nevertheless, its technical contribution should be improved and needs to be more clearly represented. The theoretical analysis should be more rigorous to reach the top-tier conference. The final recommendation is "Reject" based on the above content.

---

### Decision · Program_Chairs · 2025-01-22

Reject